**Field-scale CH$_4$ emission at a sub-arctic mire with heterogeneous permafrost thaw status**
Patryk Łakomiec[1], Jutta Holst[1], Thomas Friborg[2], Patrick Crill[3], Niklas Rakos[4], Natascha Kljun[5], Per-
Ola Olsson[1], Lars Eklundh[1], Andreas Persson[1], Janne Rinne[1]
[1] Department of Physical Geography and Ecosystem Science, Lund University, 223 62, Sweden
[2] Department of Geosciences and Natural Resource Management, University of Copenhagen,
1165, Denmark
[3] Department of Geological Sciences and Bolin Centre for Climate Research, Stockholm
University, 114 19, Sweden
[4] Abisko Scientific Research Station, Swedish Polar Research Secretariat, Abisko, 981 07, Sweden
[5] Centre for Environmental and Climate Science, Lund University, 223 62, Sweden
*Correspondence to*: Patryk Łakomiec ([patryk.lakomiec@nateko.lu.se](mailto:patryk.lakomiec@nateko.lu.se))
Abstract
The Artic is exposed to even faster temperature changes than most other areas on Earth.
Constantly increasing temperature will lead to thawing permafrost and changes in the methane
(CH$_4$) emissions from wetlands. One of the places exposed to those changes is the Abisko-
Stordalen Mire in northern Sweden, where climate and vegetation studies have been conducted
since the 1970s.
In our study, we analyzed field-scale methane emissions measured by the eddy covariance
method at Abisko-Stordalen Mire for three years (2014-2016). The site is a subarctic mire mosaic
of palsas, thawing palsas, fully thawed fens, and open water bodies. A bimodal wind pattern
prevalent at the site provides an ideal opportunity to measure mire patches with different
permafrost status with one flux measurement system. The flux footprint for westerly winds is
dominated by elevated palsa plateaus, while the footprint is almost equally distributed between
palsas and thawing bog-like areas for easterly winds. As these patches are exposed to the same
climatic and weather conditions, we analyzed the differences in the responses of their methane
emission for environmental parameters.
The methane fluxes followed a similar annual cycle over the three study years, with a gentle rise
during spring and a decrease during autumn, without emission burst at either end of the ice-free
season. The peak emission during the ice-free season differed significantly for the mire with two
permafrost status: the palsa mire emitted 19 mg-C m$^{-2}$ d$^{-1}$ and the thawing wet sector 40 mg-C m$^{-2}$
$^2$ d$^{-1}$. Factors controlling the methane emission were analyzed using generalized linear models.
The main driver for methane fluxes was peat temperature for both wind sectors. Soil water
content above the water table emerged as an explanatory variable for the three years for western
sectors and the year 2016 in the eastern sector. The water table level showed a significant
correlation with methane emission for the year 2016 as well. Gross primary production, however,
did not show a significant correlation with methane emissions.
Annual methane emissions were estimated based on four different gap-filing methods. The
different methods generally resulted in very similar annual emissions. The mean annual emission
based on all models was 3.1 ± 0.3 g-C m$^{-2}$ a$^{-1}$ for the western sector and 5.5 ± 0.5 g-C m$^{-2}$ a$^{-1}$ for
the eastern sector. The average annual emissions, derived from these data and a footprint
climatology, were 2.7 ± 0.5 g-C m$^{-2}$ a$^{-1}$ and 8.2 ± 1.5 g-C m$^{-2}$ a$^{-1}$ for the palsa and thawing surfaces,
respectively. Winter fluxes were relatively high, contributing 27 - 45 % to the annual emissions.

## 46 1 Introduction

After a period of stabilization in the late 1990s to early 2000s, atmospheric methane (CH$_4$)
concentration is increasing again at rates similar to those before 1993, which is approximately
12 ppb yr$^{-1}$ (Dlugokencky et al. 2011, Nisbet et al. 2014, Saunois 2020). The reasons behind this
increase are still partly unclear, as the mechanisms that control the global CH$_4$ budget are not
completely understood (Kirschke et al. 2013, Saunois et al. 2020). The largest natural source of
CH$_4$ are wetlands, based on top-down emission estimates (Saunois et al. 2020), and this source
may become stronger in the warming climate (Zhang et al. 2017). The shift in the isotopic
composition of CH$_4$ towards more negative values also supports the hypothesis of changes in the
biological source strength driving the increase in CH$_4$ concentration, as atmospheric CH$_4$ is
becoming more $^{13}$C-depleted (Nisbet et al. 2016).
Increasing temperature has shown to speed up the degradation of permafrost which leads to
losses in the soil carbon pool, often in the form of carbon dioxide (CO$_2$) and CH$_4$ (Malmer et al.
2005). The high northern latitudes are experiencing the fastest temperature increase due to the
ongoing global warming. Temperature changes in the Arctic have been twice as high as the global
average (Post et al. 2019).
Ecosystems near the annual near-surface air temperature isotherms of 0 °C are vulnerable to
permafrost thaw and changes in ecosystem characteristics in a warming climate. These
vulnerable ecosystems include palsa mires, such as Stordalen Mire near Abisko, Sweden, where
the recent warming has led to annual average temperatures exceeding 0 °C since 1980s
(Callaghan et al. 2010, Callaghan et al. 2013, Post et al. 2019, Figure S1). The warming has led to
an acceleration of permafrost thaw processes and a transition from palsa plateaus, underlain by
permafrost, to non-permafrost fen systems (Malmer et al. 2005). These deviations are likely to
induce changes in biogeochemical processes, including increased CH$_4$ emissions (Christensen et
al. 2003).
The most direct micrometeorological field-scale method used to measure CH$_4$ exchange between
ecosystem and atmosphere is the eddy covariance (EC) method (e.g. Verma et al. 1986, Aubinet
et al., 2012). The advantages of this method are its high temporal resolution and minimal
disturbance to the measured surface. Thus, it is feasible for long-term measurements of rates of
gas exchange that integrates over surface variation (Knox et al. 2016, Li et al. 2016, Rinne et al.
2018). However, information on the small-scale spatial distribution of surface fluxes is lost with
the method due to the spatially integrative nature of the EC method. Instead of resolving the
small-scale spatial variability, the EC method provides averaged fluxes from a larger area, the flux
footprint area (Kljun et al. 2002). However, spatial variability can be resolved by the EC method
using measurements conducted under different wind directions, as the footprint area is located
upwind of the measurement tower. We can take advantage of this feature to obtain gas exchange
rates from two different ecosystem types with one measurement system by placing the
measurement system on the border between these systems (e.g. Jackowicz-Korczyński et al.,
2010; Kowalska et al., 2013; Jammet et al., 2015; 2017). Stordalen Mire offers an excellent
opportunity to conduct flux studies where one flux system is used to monitor two ecosystem
types since the wind direction is bimodal. While previous studies in the area have compared open
water surfaces to completely thawed fen (Jammet et al., 2015, 2017, Jansen et al. 2020), no
comparison of field-scale $CH_4$ emission between permafrost palsa plateaus and thawing wet
areas has been conducted yet.
Previous studies on $CH_4$ emission within the Stordalen Mire from areas with different permafrost
status have been done using chamber measurements (McCalley et al. 2014, Deng et al. 2014).
McCalley et al. (2014) reported $CH_4$ emissions from palsas underlain by permafrost to be close to
zero, summertime emissions from thawing wet areas to be around 25 mg-C $m^{-2}$ $d^{-2}$, while
completely thawed fen sites revealed much higher emission of 150 mg-C $m^{-2}$ $d^{-2}$. There are only
few wintertime data on $CH_4$ emission available using the chamber method (Christensen et al.
2000, Nilsson et al. 2008, Godin et al. 2012, McCalley et al. 2014). However, EC measurements
conducted at different northern mires typically show low but positive emissions in winter (Rinne
et al., 2007; Yamulki et al. 2013, and others).
In this study we analyzed field-scale $CH_4$ emission from two areas of Stordalen subarctic mire.
The first area is dominated by drained permafrost plateau. The second area is thawing and thus
resulting in wetter conditions. Outputs from this analysis are differences in the $CH_4$ emissions
from the mire patches with heterogeneous permafrost status. We are expecting, based on the
previous studies, that fluxes from the wetter sector will be around 30 mg-C $m^{-2}$ $d^{-2}$, while the
palsa plateau will emit significantly lower fluxes during the peak season. We presume that
winter fluxes will be positive but very low.
For estimation of annual $CH_4$ emission we need gap-free datasets. Up to date, there is no
generally accepted gap-filling method for $CH_4$ fluxes, hence four different gap-filling methods
were compared. The test of the four methods will decrease the uncertainty in the annual balance
estimation (Hommeltenberg et al. (2014), Rößger et al. (2019), Kim et al. (2019)). It was important
to use more than one method in this case of study because datasets were portioned and due to
that contained more gaps.
This study aimed to estimate the annual $CH_4$ emission from two distinct different ecotypes, with
heterogeneous permafrost status, exposed to the same environmental factors.  Furthermore, we

analyzed the seasonal cycle of $CH_4$ emission to quantify the contribution during different seasons. Moreover, an analysis of differences in controlling factors for these two different areas was done.

# 2 Materials and method

## 2.1 Study site

The study area is Stordalen Mire, a mire complex underlain by discontinuous permafrost located in northern subarctic Sweden (68°20' N, 19°30' E) near Abisko (Ábeskovvu). The station Abisko-Stordalen (SE-Sto) is a part of the ICOS Sweden research infrastructure and is the only one in Sweden situated in the subarctic region. The measurement period that is analyzed here covers three years from 2014 to 2016. The mean annual near- surface air temperature in this region has been increasing during the last decades, and temperatures recorded by SMHI (Sveriges meteorologiska och hydrologiska institut) at ANS (Abisko Naturvetenskapliga Station) has exceeded the 0 °C threshold since the late 1980s (Callaghan et al. 2013, Figure S1). During the years 2014-2016, the mean near-surface air temperature (Ta) was 1.0 °C and 0.3 °C at ANS and the ICOS Sweden station Abisko-Stordalen (SE-Sto), respectively. The average annual precipitation, based on ANS data, is around 330 mm yr$^{-1}$. An acceleration of permafrost loss with increasing temperatures is likely (Callaghan et al. 2013).

The large mountain valley of Lake Torneträsk (Duortnosjávri) channels winds at the study site, leading to a bimodal wind distribution (Figure 1), which allows us to divide our analyses into two distinct sectors. The plant community structure around the tower is determined by the hydrology which in turn is determined by the microtopographic variation in the surface due to the local permafrost dynamics. Different plant communities would have different productivities thus controlling the $CO_2$ and $CH_4$ fluxes from those surfaces. The area to the west of the EC mast is dominated by a drier permafrost palsa plateau hereafter referred to as the western sector, whereas the area to the east is a mixture of thawing wet areas and palsas, hereafter referred to as the eastern sector. The drained permafrost plateau is dominated by *Empetrum hermaphroditum*, *Betula nana*, *Rubus chamaemorus*, *Eriophorum vaginatum*, *Dicranum elongatum*, *Sphagnum fuscum*. The wet areas are characterizing by *E. vaginatum*, *Carex rotundata*, *S. balticum*, *Drepanucladus schulzei*, *Politrichum jensenii* (Johansson et al. 2006). The thawing areas in this sector exhibit ombrotrophic, bog-like, features. Dominant vegetation varies with the microforms of the mire.

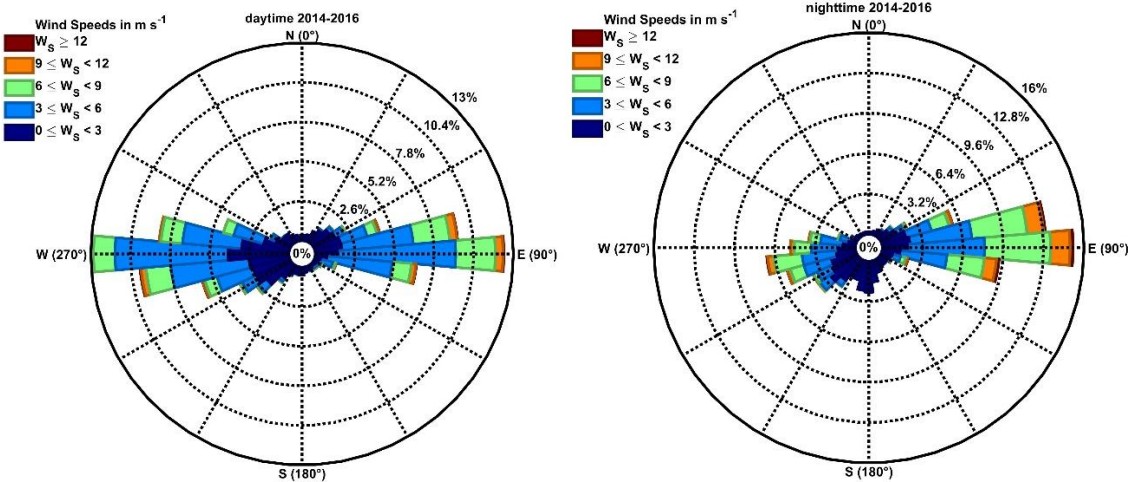


Figure 1. The wind rose for SE-Sto tower for years 2014-2016 for the daytime (left panel) and nighttime
(right panel)


## 2.2 Flux measurements

The EC measurements of $CH_4$ fluxes at SE-Sto are made using a closed-path fast off-axis
integrated cavity output spectrometer (OA-ICOS LGR model GGA-24EP, ABB Ltd, Zurich,
Switzerland) combined with a 3-D sonic anemometer (SA-Metek uSonic-3 CLASS A, Metek GmbH,
Germany). Air was sampled via a 29.6 m long polyethylene tubing with an 8.13 mm inner
diameter. Analysis of the high-frequency loss were performed to assess the effect of relatively
long sample tubing. We analyzed this with the co-spectra of the $CH_4$ and the vertical wind speed
w. The analysis did not show a dampening effect at the high frequencies (Figure S2), thus the high
frequency attenuation does not seem to be very large. Furthermore, the post-processing
software we used to calculate fluxes includes correction for high-frequency losses. The nominal
tube flow rate was 36 l min$^{-1}$. The sampling inlet was displaced 22 cm horizontally of the sonic
anemometer measurement volume towards 180°. The response time of the LGR-FGGA was 0.1 s.
The LGR FGGA was placed inside a heated and air-conditioned shelter. The anemometer was
located north of the instrument shelter and was oriented with the sensors north pointing towards
186°. This orientation allows undisturbed wind measurements from both main wind directions,
East and West.
$CO_2$ and $H_2O$ were measured with a LI-COR LI-7200 (LI-COR Environment, USA) closed path infra-
red gas analyzer. The sampling inlet was at the same location as the sampling point for the $CH_4$
analyzer. Sampled air was transported through 1.05 m and of 5.3 mm ID tubing. The nominal
tube flow rate was 15 l min$^{-1}$.
The anemometer and air sampling tubes were mounted on a mast of 2.2 m above ground level
(a.g.l.) (68°21'21.32" N, 19°2'42.75" E), placed at the edge of the western and the eastern sectors.
Data were collected by an ISDL data logger (In Situ Instrument AB, Sweden) with a 20 Hz time
resolution.

## 2.3 Ancillary Measurements
Ancillary measurements are presented in Table S1. The sampling frequency for these parameters
was 1 Hz and the collected data were averaged into half-hourly values. Measured variables are
divided into two categories: peat/soil parameters, and meteorological parameters. Peat
temperatures at each depth, soil heat fluxes, and soil water contents (SWC) were measured at
four locations around the EC tower, located towards the four cardinal directions. In further
analysis, data just from two of these locations were used (East and West) as these were within
the flux footprint areas of the EC tower. The sites for the water table level (WTL) measurements
differed from the peat temperature profiles. The soil pit for temperature and moisture probe in
the western sector is located on a palsa plateau. However, the WTL probe is located in a pond
approximately 10 m away from the soil temperature and SWC measurement, as there is no WTL
above the permafrost of palsas. The soil pit for temperature and SWC probe in the eastern sector
is located in the wet thawing area. The WTL probe is located in the wetter area approximately 10
m away. Furthermore, data for WTL was available only during the unfrozen period, as the probes
were removed during the frozen period to avoid damage. Meteorological variables were
measured on a separate mast, placed 10 meters south-west of the flux measurement mast.

## 2.4 Flux calculation
Fluxes of $CO_2$, $CH_4$, $H_2O$, and sensible heat were calculated using EddyPro 6.2.1 (LI-COR
Environment, USA) as half-hourly averages. The data quality flagging system and advanced
options for EddyPro were set up following Jammet et al. (2017). The wind vector was rotated by
a double rotation method and data were averaged by block averaging (Aubinet et al. 2012). The
time lag was obtained by maximizing the covariance (Aubinet et al. 2012).
Based on the wind direction, the half-hourly data were divided into western and eastern datasets,
similarly to analyses by Jackowicz-Korczyński et al. (2010) and Jammet et al. (2015, 2017). The
eastern dataset contained fluxes and other variables recorded when the wind was from 45°-135°,
and the western dataset parameters when wind directions were 225°-315°. These two datasets
were analyzed separately. Fluxes measured with wind from these two sectors are influenced by
mire surfaces dominated by differing permafrost status, moisture regimes, and plant community
structures. These reflect the thaw stages of a dynamic arctic land surface, responding to the
warming climate. These two wind sectors include more than 80 % of all data during the years
2014-2016. Northerly and Southerly wind directions, i.e. winds from outside these sectors
occurred mainly in low wind speed conditions. The distribution of wind directions is presented in
Figure 1.
CH$_4$ fluxes were filtered by quality flags according to Mauder and Foken (2004). These indicate
the quality of measured fluxes, "0" being the best quality fluxes, "1" being usable for annual
budgets, and "2" being flux values that should not be used for any analysis. Thus, in further
analysis fluxes with flag "2" were removed. Also, consecutive data points originating from the two
pre-defined wind direction sectors were removed to avoid influences from non-stationary conditions.
We also analyzed the behavior of the CH$_4$ fluxes against low turbulence conditions using friction
velocity (u*) as a measure of turbulence. We binned the CH$_4$ fluxes into 0.05 m s$^{-1}$ u* bins and
plotted the binned CH$_4$ flux values against u* in 40-day windows over the growing period (d.o.y.
150-250, d.o.y. 210 was the beginning of the last averaging window). The CH$_4$ flux showed no
dependence on u* below 0.6 m s$^{-1}$. A slight positive correlation was found during stronger
turbulent conditions (u* > 0.6 m s$^{-1}$), but we deemed this not high enough to warrant exclusion
of those points from further analysis. Thus, we did not remove data based on the results of u*.
The fraction of data remaining, after filtering based on the quality flags and other criteria
described above, is presented in Table 2.
The analysis of relations of CH$_4$ fluxes to environmental parameters was done using the non-gap-
filled dataset of daily averages, to avoid the danger of circular reasoning of analyzing the relations
to the same factors that were used for gap-filling.

## 2.5 Footprint modeling and land cover classification

A detailed land cover classification was performed for the EC-tower footprint area to estimate
the flux contribution from the drained palsa and the thawing wet areas. We used images over
the Stordalen Mire collected with an eBee (SenseFly, Lausanne, Switzerland) Unmanned Aerial
Vehicle (UAV) carrying a Parrot Sequoia camera (Parrot Drone SAS, Paris, France) on July 31, 2018.
The images were processed in Agisoft Photoscan (Agisoft LLC, St. Petersburg, Russia) to create an
orthomosaic and a Digital Surface Model (DSM) with spatial resolutions of 50 cm x 50 cm. Field
data for training a classification were collected in mid-August 2018 with sampling areas of 50 cm
× 50 cm that were classified into wet or dry, and a random forest classification was performed to
classify the footprint into wet and dry areas with the orthomosaic and DSM as input. The dry
areas in the flux footprint areas of SE-Sto footprint correspond to palsas, while the wet areas are
thawing surfaces.
Flux footprints were calculated with the FFP model (Kljun et al. 2015). Receptor height, Obukhov
length, standard deviation of lateral velocity fluctuations, friction velocity, and roughness length
were used as input data. The input data were divided into the two wind sectors mentioned above,
before footprint calculation, and footprints were calculated separately for them. We calculated
footprints for each half-hourly data point and aggregated these to annual footprint climatologies
for each sector separately.  I.e. the half-hourly footprint function values were aggregated for each
land cover grid cell (50 cm x 50 cm) to derive a footprint-weighted flux contribution per pixel.
Based on the land cover classification and annual CH$_4$ fluxes for each sector, combined and
weighted with the footprint climatology, it was possible to estimate annual emissions from the
different surface type.

## 2.6 Gap-filling methods for CH$_4$

We compared four different gap-filling methods, separately for both sectors. These methods
were: look-up tables (REddyProc ("Jena gap-filling tool"), Wutzler et al. 2018), 5-day moving
mean, artificial neural network (Jammet et al. 2015, 2017), and generalized linear models (Rinne
et al. 2018). All these methods, except for moving mean, have been used before for gap-filling
CH$_4$ flux data from different mire ecosystems. The look-up table approach uses half-hourly data,
while for the other three methods we used daily average data, as CH$_4$ emissions from this
ecosystem do not show a diel cycle (see below, Section 3.2 for a detailed description).
The uncertainties due to each method were analyzed by the introduction of artificial gaps to the
data, with lengths comparable to gaps existing in the year 2014. 35-day and 80-day gaps were
implemented to the data of years 2015 and 2016. Gaps were placed in the winter period, to
obtain similar gap distribution as in the year 2014 (gap distribution is presented below in Table
3). Annual sums, with artificial gaps, were compared with results from methods without those
gaps. Statistical significances of differences between models were analyzed by using a two-
sample t-Test for equal means with a 95 % confidence level (MATLAB R2019b).

### 2.6.1 REddyProc

The Jena gap-filling tool using look-up tables requires half-hourly data of CH$_4$ flux and
environmental data: shortwave incoming radiation, air temperature, soil temperature, relative
humidity, and friction velocity. Based on environmental data, fluxes are classified and averaged
within a given time window. The missing data are then filled with the average value from
classified data. Uncertainty can be estimated as standard deviations of fluxes within classes.
Detailed information about the method is presented by Falge et al. (2001) and Wutzler et al.
275 (2018).


### 2.6.2 Moving average

A 5-day moving mean approach is a very simple gap-filling method where the moving mean is
calculated for subsets of the data. In case of a gap in the averaging window, the mean value is
calculated for fewer observations. The method was applied on daily average CH$_4$ flux data using
MATLAB (movmean function).  For gaps longer than 5 days, linear interpolation was used
between the last point before the gap and the first point after gap. Uncertainties of the single
gap-filled flux were estimated by calculating the moving standard deviation (movstd function,
MATLAB) on the same subset of the data like for the moving mean.

### 2.6.3 Artificial Neural Network

An artificial neural network (ANN) has been successfully applied for gap-filling of $CH_4$ fluxes by e.g. Dengel et al. (2013), Jammet et al. (2015,2017), Knox et al. (2016) *and* Rößger et al. (2019). This type of ANN was designed in MATLAB using a fitnet function with 30 hidden neurons. We used the Levenberg-Marquardt algorithm as a training function (Levenberg 1944 Marquardt 1963). All available daily average $CH_4$ values were used to train (70 %), validate (15 %), or test (15 %) the ANN. The ANN requires input data without gaps to work properly and thus the short gaps (up to three days) in environmental daily averaged data were filled by linear interpolation before the ANN analysis. All environmental variables, except the WTL were used as input for the ANN method. The WTL was excluded because it was not available during the frozen period, i.e. most of the year. The ANN method was applied to sectors and each year separately (ANN YbY) or all three years together. Multiple repetitions were done to minimize uncertainty connected with randomly chosen data points for training, validation, and testing. The network was trained and used to calculate the time series of $CH_4$ daily fluxes 100 times in each case of gap-filling. The number of repetitions was chosen to have a sample large enough to calculate reliable mean and standard deviation values, and to keep the computation time reasonably short. An average $CH_4$ flux for each day was calculated based on 100 daily values. The gaps in the measured flux time series were filled with values from the time series calculated by ANN. Errors were estimated as standard errors of mean on daily flux, based on 100 ANN trained values.

### 2.6.4 Generalized Linear Model

Generalized linear models (GLM) are linear combinations of linear and quadratic functions describing the dependence of response variables to predictors. In our case, the response variable was the logarithm of daily average $CH_4$ flux, and predictors were daily averages of measured environmental variables. Controlling factors of $CH_4$ emission were examined by a procedure similar to the routine described by Rinne et al. (2018). A correlation matrix of linear correlation based on daily values of environmental factors and $CH_4$ fluxes was constructed (Figure S3). Additionally, the logarithm of $CH_4$ fluxes was added to the correlation matrix to check the exponential relationship between parameters. This type of relationship between $CH_4$ fluxes and peat temperature was previously found by e.g. Christensen et al. (2003), Jackowicz-Korczyński et al. (2010), Bansal et al. (2016), Pugh et al. (2017) and Rinne et al. (2018). Gap-filled $CO_2$ flux, and gross primary production (GPP), were also included as prospective controlling factors. In order to avoid strong cross-correlation between predictors, first, we selected the parameter with the highest correlation and then removed parameters from the GLM development with a cross-correlation between parameters $R^2 > 0.6$. We thus chose GPP, soil temperature at 30 cm depth for the eastern sector and 10 cm depth for the western sector, soil water content (SWC), short-wave incoming radiation, and vapor pressure deficit (VPD) as possible predictors. The model was constructed in MATLAB using the stepwiseglm function (Dobson 2002). The GLM was made

separately for each year (GLM YbY) and for all three years combined. Errors were estimated as
95 % confidence intervals because it was an output of the stepwise function. This method was
also used for the determination of the controlling factors from the possible predictors.

## 2.7 Gap-filling of $CO_2$ fluxes

$CO_2$ fluxes were calculated for both wind sectors. $CO_2$ flux exhibited a diel pattern in the growing
season, with uptake during daytime (shortwave incoming radiation > 50 W m$^{-2}$) and release at
night (shortwave incoming radiation < 50 W m$^{-2}$). We used the ANN to gap-fill the time series of
$CO_2$ fluxes. This method was chosen to check the possibility to reconstruct the diel cycle. This diel
pattern of $CO_2$ was taken into account by using half-hourly data. We used all environmental
variables excluding the WTL, as for $CH_4$ fluxes. GPP was obtained by partitioning the gap-filled
data using the Jena gap-filling tool. Finally, the half-hourly gap-filled GPP and $CO_2$ data were
averaged to daily values.

## 2.8 Contribution of palsa and thaw surfaces to average $CH_4$ emission

Using the average annual $CH_4$ emission from the two wind sectors and the relative contributions
of the two surface types to the fluxes from these sectors, we calculated the average annual
emission from these surface types. We expressed the average annual $CH_4$ fluxes for the two
sectors, $F_e$ (East) and $F_w$ (West), with a pair of equations,
$$F_e = f_{e,p}E_p + f_{e,t}E_t, \qquad\qquad\qquad\qquad\qquad (1)$$
$$F_w = f_{w,p}E_p + f_{w,t}E_t, \qquad\qquad\qquad\qquad\qquad (2)$$
where $f$ indicates the fractional contribution of surface type to the flux from the footprint
calculations (subscripts $e$ and $w$ referring to east and west, respectively; $p$ and $t$ to palsa and thaw
surface, respectively); and $E_p$ and $E_t$ are emissions from palsa and thaw surface, respectively. We
solved this equation set with two unknowns to yield $E_p$ and $E_t$. Here we assumed that the emission
rate from both palsa and thaw surfaces are equal in eastern and western sectors. Furthermore,
we must assume that there is no correlation between footprint contribution and seasonally
developing emission rate at either surface type. The seasonally constant contributions of the
surface types to the footprint indicate that the latter assumption may well be valid (Figure S4).

## 2.9 Definition of seasons

The beginning of the unfrozen period was defined as the day when daily averages of peat
temperature at 10 cm depth had been above 0 °C for three consecutive days. The end of the
unfrozen period was defined as the day when daily averages of peat temperature at 10 cm depth
had been below 0 °C for three consecutive days. The unfrozen and frozen periods commence in
the western sector on average 3 days earlier than in the eastern sector, but differences in the
unfrozen season length are not systematic (Figure 2). The beginning and the end of the unfrozen

season were determined independently for both sectors. The horizontal distance between soil
temperature sensors in eastern and western sectors was around 75 m, differed about 2 m in
elevation, and the distance from the flux tower was roughly 40 m.

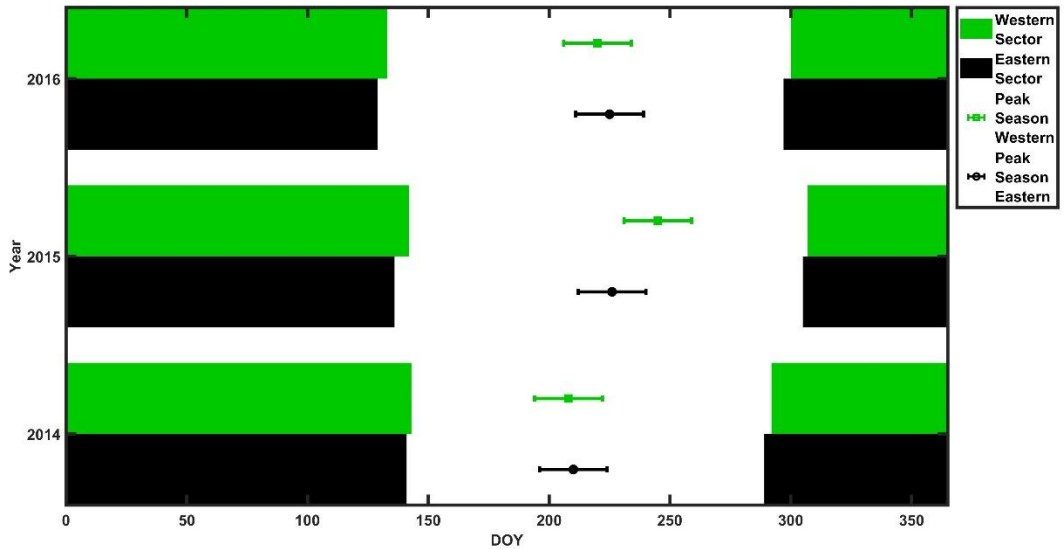

Figure 2. Time periods of frozen peat during the years 2014 - 2016 (green and black bars) and
peak $CH_4$ emission season (dot with whiskers) for the western sector (green) and the eastern
sector (black). (For peak season definition see Section 3.2)

## 3 Results

### 3.1 Environmental conditions and flux footprints

Winds from eastern and western sectors contributed to 50 % and 40 % to the daytime wind
directions, respectively (Figure 1). Northerly and southerly winds contributed to around 5 % each.
In the nighttime, 51 % of wind was from the East and 32 % from the West. Additionally, 15 % of
total wind came from the South during nighttime, probably as catabatic flow from higher
mountain areas. The wind from North was rare, around 2 % of all the cases.

The annual average peat temperature of the uppermost 50 cm of peat was systematically warmer
in the eastern sector than in the western sector (Table 1; Figure 3). However, the summertime
peat temperature at the top 10 cm layer was warmer for the western sector (Figure S5). The
situation was the opposite during winter when the western sector down to 50 cm was colder
than the eastern sector. During our investigation period (2014-2016), the peat temperatures
from 30 cm to 50 cm below ground were colder in the western sector than those of the eastern
sector, corresponding to the existence of the permafrost. Temperature differences, between
both areas, at the same depth, were stable over the measurement years. The biggest difference

was noticed at a depth of 30 cm. The temperatures at 30 cm and 50 cm depth were increasing
during consecutive years.


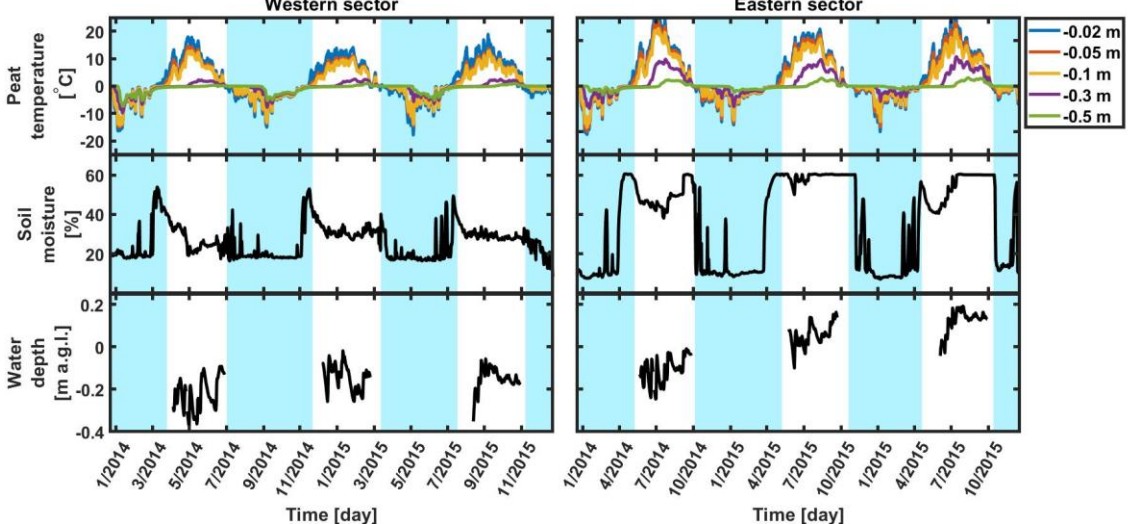

Figure 3. Time series of daily mean values for western and eastern sectors for: peat temperature (top
panel), soil moisture (middle panel), and water table level (bottom panel), where the shaded light blue
area is the frozen period, when peat temperature at 10 cm was below 0 °C (see Section 2.8 for a detailed
description).

Table 1. Mean annual air and peat temperatures for the years 2014-2016 for eastern and western
sectors.

| | Temperature [°C] | | | | | | | | |
|---|---|---|---|---|---|---|---|---|---|
| depth in cm | 2014 E | 2014 W | 2014 E-W difference | 2015 E | 2015 W | 2015 E-W difference | 2016 E | 2016 W | 2016 E-W difference |
| ambient air | 0.3 | 0.3 | - | 0.1 | 0.1 | - | 0.3 | 0.3 | - |
| 2 | 1.6 | 1.4 | 0.2 | 2.2 | 2.0 | 0.2 | 2.2 | 1.9 | 0.2 |
| 5 | 1.4 | 0.8 | 0.5 | 1.9 | 1.3 | 0.7 | 1.9 | 1.3 | 0.6 |
| 10 | 1.2 | 0.5 | 0.6 | 1.7 | 1.1 | 0.7 | 1.7 | 1.1 | 0.6 |
| 30 | 0.3 | -0.9 | 1.2 | 0.6 | -0.6 | 1.2 | 0.8 | -0.5 | 1.3 |
| 50 | -0.1 | -1.0 | 0.8 | 0.0 | -0.8 | 0.8 | 0.2 | -0.6 | 0.8 |


The WTL was higher in 2014 than in 2015 and 2016 according to measurements both in the
eastern and western sectors (Figure 3). This is not reflected in the SWC measurements, which is
probably due to the different locations of the measurements of WTL and SWC. In the western
sector the WTL was measured in an isolated wet patch, surrounded by drier palsa and thus it is
not representative of the dominating type of this area. The WTL in the eastern sector was more
representative of the area of the footprint. Data from the WTL probe in the West part of the mire
was excluded from the further analysis as it does not represent the situation for the majority of
the western sector. The soil moisture was higher for the eastern than the western sector during
all years. The data shows a distinctive step change at thaw and freeze, as the dielectricity of ice
and liquid water differ. In the eastern sector, the soil was fully saturated for most of the unfrozen
period during the years 2015-2016, while 2014 indicates lower water content levels. The western
sector was never fully saturated at any time during the years 2014-2016.
Footprint and flux contribution of drier and wetter areas are presented in Figure 3. The dry areas
(yellow) contribute on average over all three years to more than 90% of the fluxes measured
from the western sector at the eddy covariance tower. In the eastern sector, the wetter (blue)
and drier areas contribute almost equally to the fluxes. The contributions of the wet and dry
areas to the fluxes in both sectors remained almost constant across the three study years.

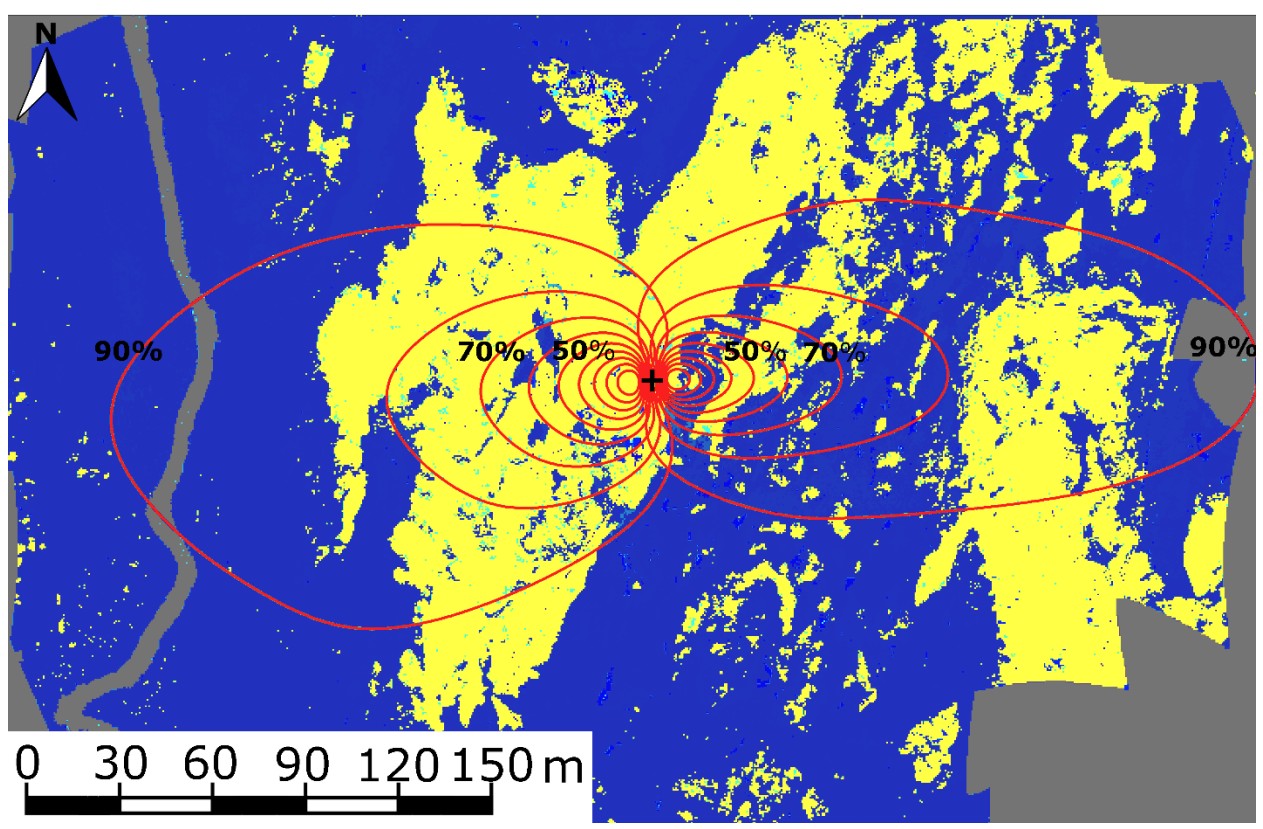


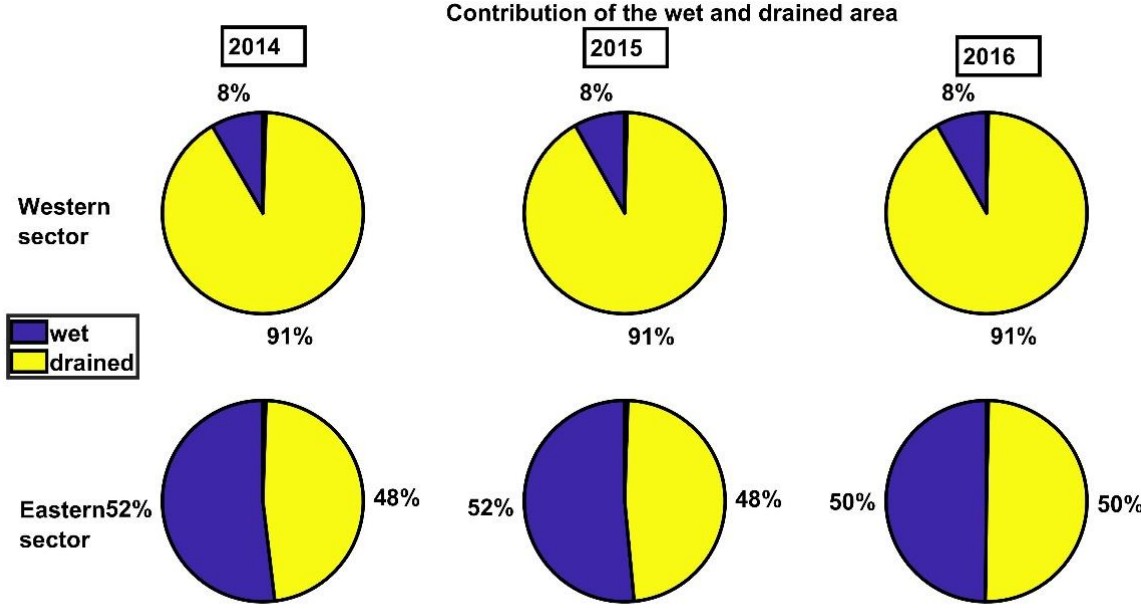


Figure 4. Footprint-weighted contribution of the wet and drained area at the SE-Sto tower (upper panel)
for the year 2014 and relative amounts of wetter areas (blue) and drained palsa area (yellow) inside the
80 % area of influence of the footprints (lower panel). The black cross is the location of the tower and
each red line indicates 10 % of the contribution from the source area to measured fluxes at the tower.
The footprint climatology is almost identical for all study years, see bottom panel.

422

### 3.2 CH$_4$ fluxes

We analyzed the growing season data of each year and both wind sectors separately in regards
to a possible diel cycle of CH$_4$ fluxes. This was done by normalizing each half-hourly flux by
dividing it with the daily median from that day for the whole growing season (Rinne et al. 2007).
This yielded a normalized diel cycle of CH$_4$ fluxes. Figure S6 shows slightly lower emission during
mid-day hours. However, the difference is small compared to the short term variation in the
fluxes as indicated by the interquartile range. Thus, for the purpose of gap-filling this effect could
be negligible in calculating daily averages. However, it is interesting to observe this type of diel
cycle, with minima at daytime. It could be linked to the temperature cycle of the top peat layer.
This could affect the methanotrophy, while the methanogenesis occurring at slightly deeper
layers would be less affected. This would lead to higher methanotrophy at daytime and thus
lower emission. It is possible to calculate CH$_4$ daily averages without gap-filling the diel cycle,
similarly to e.g. Rinne et al., (2007, 2018) and Jackowicz-Korczyński et al. (2010). We discarded
daily averages with less than 10 flux data points from further analysis, to ensure the reliability of
the daily average fluxes. Uncertainties of daily averages were calculated as standard errors of the
mean. The size of the available flux dataset, after gap-filling by daily averaging, is presented in
Table 2. The gap distribution in the datasets for the different sectors and years is presented in
Table 3.

441

Table 2. The size of available daily data sets after gap-filling by daily averaging for each year and wind sector.

|  | 2014 E | 2015 E | 2016 E | 2014 W | 2015 W | 2016 W |
|---|---|---|---|---|---|---|
| total number of points | *365* | *365* | *366* | *365* | *365* | *366* |
| number of points after averaging | *137* | *174* | *182* | *96* | *167* | *178* |
| % of available data | *38* | *48* | *50* | *26* | *46* | *49* |
| % of available data during winter period | *36* | *54* | *56* | *12* | *36* | *37* |
| % of available data during unfrozen period | *40* | *41* | *42* | *47* | *58* | *63* |

444

Table 3. Gaps distribution over years and wind direction.

| Type of gap | Length of gap | 2014 E | 2015 E | 2016 E | 2014 W | 2015 W | 2016 W |
|---|---|---|---|---|---|---|---|
| short gap | 1-3 day | *32* | *50* | *41* | *24* | *44* | *36* |
| medium gap | 4-7 day | *7* | *12* | *11* | *6* | *8* | *11* |
| long gap | 8-30 day | *3* | *7* | *4* | *4* | *6* | *6* |
| very long gap | > 30 day | *1* | *0* | *0* | *3* | *0* | *0* |

446

Daily non-gap-filled $CH_4$ fluxes showed a characteristic annual cycle, with peak emissions in August (Figure 5) and low but positive wintertime fluxes. Wilcoxon rank sum test need data without autocorrelation. The autocorrelation in the data existed up to 8 days. Based on this we divided winter data with subsets where every 9th day was selected. We tested the difference of those subsets to zero with Wilcoxon rank sum test. Winter fluxes were statistically different from zero ($p < 0.001$, two-sided Wilcoxon rank sum test). Winter fluxes from the western and eastern sectors were also different from each other ($p < 0.001$).

$CH_4$ fluxes, both from the western sector and the eastern sector started increasing after snowmelt up to a maximum in August (Figure 5). No major springtime emission burst nor autumn freeze-in burst were observed in any of the years.


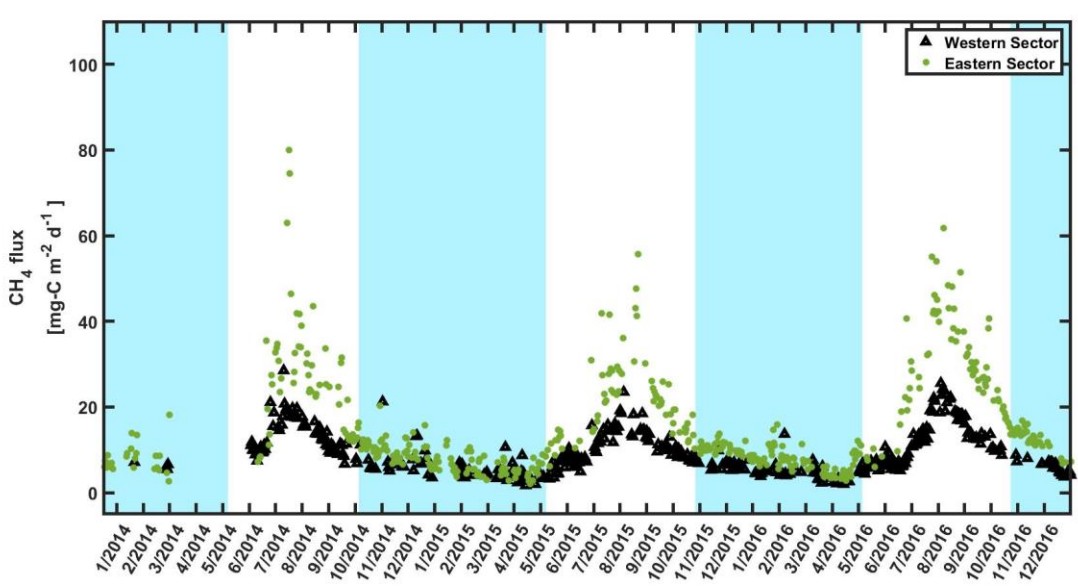


Figure 5. Time series for non-gap-filled $CH_4$ daily averaged fluxes for the western sector (green triangles)
and the eastern sector (black dots), where the shaded light blue area is frozen period when peat
temperature at 10 cm was below 0 °C (see Section 2.8 for a detailed description).

The middle-day of the peak season of the $CH_4$ emission was defined as the maximum of the 14-
days moving average. Two weeks forward and backward from the middle-day was defined as the
peaks season and emissions were estimated for that period in each year. The average emission
during the peak seasons was 40 mg-C m$^{-2}$ d$^{-1}$ for the eastern sector and 19 mg-C m$^{-2}$ d$^{-1}$ for the
western sector. Detailed emissions for all years are presented in Table 4. The peak season
emissions were statistically different from each other (p < 0.001). Wintertime fluxes were steadily
declining as winter continued and the lowest emissions were observed slightly before the spring
thaw. Wintertime average emissions were 9 mg-C m$^{-2}$ d$^{-1}$ for the eastern sector and 6 mg-C m$^{-2}$
d$^{-1}$ for the western sector. Detailed emissions of winter periods are presented in Table 5.
Table 4. $CH_4$ emission during the peak season

|  | Mean | Standard deviation | The standard error of the mean |
|---|---|---|---|
|  | [mg-C m$^{-2}$ d$^{-1}$] | | |
| 2014 E | 40.7 | 17.2 | 4.3 |
| 2015 E | 34.4 | 11.7 | 3.7 |
| 2016 E | 45.4 | 6.7 | 1.7 |
| 2014 W | 18.6 | 3.2 | 0.8 |
| 2015 W | 16.1 | 3.2 | 1.0 |
| 2016 W | 20.9 | 2.6 | 0.7 |


Table 5. CH$_4$ emission during the winter period

|        | Mean | Standard deviation | The standard error of the mean |
|--------|------|--------------------|--------------------------------|
|        | [mg-C m$^{-2}$ d$^{-1}$] | | |
| 2014 E | 9.0  | 2.8  | 0.4 |
| 2015 E | 8.3  | 1.7  | 0.2 |
| 2016 E | 9.8  | 2.6  | 0.3 |
| 2014 W | 7.2  | 2.2  | 0.4 |
| 2015 W | 5.5  | 1.4  | 0.2 |
| 2016 W | 5.2  | 3.4  | 0.4 |



### 3.3 Factors controlling the CH$_4$ fluxes

In the eastern sector, the CH$_4$ flux correlated best with the peat temperature at 30 cm depth, and
in the western sector with the temperature at 10 cm depth. Using temperatures above the level
of maximum correlation led to similar hysteresis-like behavior in CH$_4$ flux - temperature relations
as presented by Chang et al. (2020), but using deeper temperatures led to inverse hysteresis
compared to shallower temperatures (Figure 6). The correlation matrix (Figure S3) shows the
importance of SWC in the CH$_4$ emissions, while WTL does not correlate significantly with CH$_4$ flux.
Controlling factors were examined before and after temperature normalization of the CH$_4$ fluxes
following Rinne et al. (2018) (Table 6), in order to avoid effect of cross-correlation between
explanatory parameters.

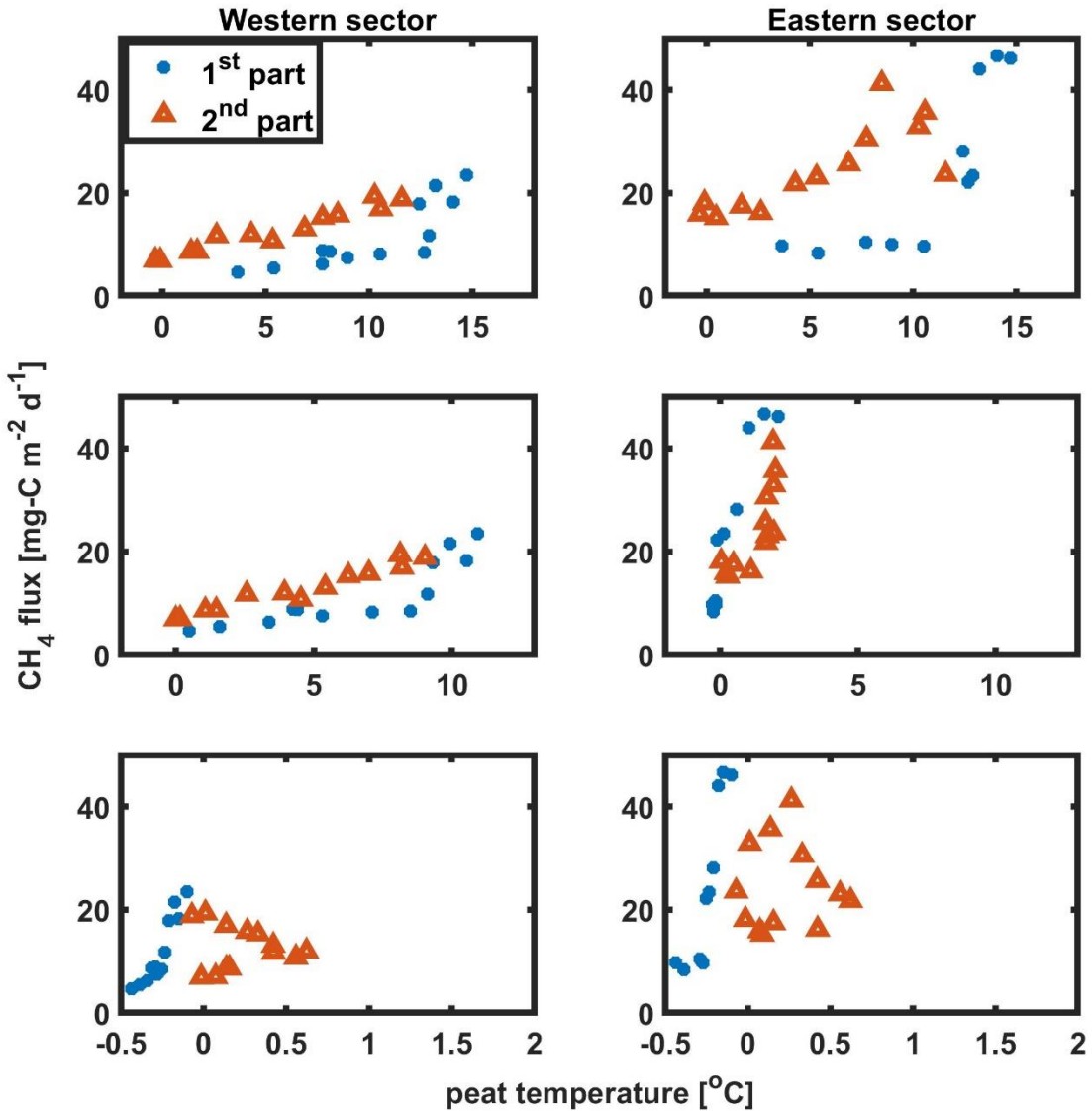

Figure 6. Weekly averages of CH4 fluxes against the surface peat temperature (top panels), the depth with best correlation (middle panels), and the deeper layer (bottom panel). Data were divided into the 1st part of the growing season (blue dots) before the maximum weekly emission, and 2nd part of the growing season (orange triangles) after that.

The result from GLM, showing the variables that contribute to the model, is presented in Table S2. The parameter that was selected first by all models, was peat temperature, at 10 cm depth for the western sector and at 30 cm depth for the eastern sector. For the eastern sector, the GLM algorithm selected SWC as the explanatory factor for $CH_4$ fluxes during all years as well as for the combined three-year period. The GLMs created for the western sector did not have other explanatory factors besides the peat temperature that were selected in all years. However, two

more explanatory factors, GPP and shortwave incoming radiation, appeared in the three time
periods (years 2015 and 2016, and three-years combined) for the western sector.
The eastern sector models had shortwave incoming radiation as the explanatory factor for the
year 2015, the year 2016, and combined three-year period. A unique variable for this sector was
the vapor pressure deficit, which was used in the models constructed for the years 2016 and
combined three-year period.
The year 2014 was characterized by a smaller number of parameters contributing to the models
for both sectors compared to other years and combined three-year models. Only peat
temperature and SWC were explanatory variables for both sectors in this year. The years 2015
and 2016 and all three years combined have a longer list of parameters.
As the WTL data was available only during a short period of the year, it was not analyzed with the
GLM. The WTL measurement in the western sector was not representative of the conditions for
most of the sector, this parameter was not used for further analysis from this sector. The WTL
was correlated with $CH_4$ fluxes for the eastern sector.
Based on the chosen explanatory variables it was noticed that the seasonal cycle could be
explained by a lower number of parameters than the interannual variation.
Table 6. Summary of controlling factors before and after temperature normalization

| Year and ecosystem | R for $CH_4$ flux | the p-value for $CH_4$ flux | R for temperature normalized $CH_4$ flux | the p-value for temperature normalized $CH_4$ flux |
|---|---|---|---|---|
| GPP | | | | |
| 2014 E | 0.71 | $7 \times 10^{-22}$ | -0.03 | 0.70 |
| 2015 E | 0.69 | $2 \times 10^{-25}$ | 0.02 | 0.83 |
| 2016 E | 0.77 | $1 \times 10^{-36}$ | 0.21 | $4 \times 10^{-3}$ |
| 2014 W | 0.69 | $4 \times 10^{-15}$ | -0.10 | 0.36 |
| 2015 W | 0.73 | $6 \times 10^{-29}$ | 0.05 | 0.56 |
| 2016 W | 0.71 | $5 \times 10^{-29}$ | -0.02 | 0.76 |
| WTL | | | | |
| 2014 E | -0.50 | $2 \times 10^{-4}$ | $1 \times 10^{-2}$ | 0.94 |
| 2015 E | -0.20 | 0.30 | -0.20 | 0.17 |
| 2016 E | 0.60 | $4 \times 10^{-6}$ | -0.30 | 0.01 |
| SWC | | | | |
| 2014 E | 0.51 | $2 \times 10^{-10}$ | -0.02 | 0.79 |
| 2015 E | 0.51 | $1 \times 10^{-12}$ | -0.03 | 0.66 |
| 2016 E | 0.69 | $1 \times 10^{-26}$ | 0.20 | $6 \times 10^{-3}$ |
| 2014 W | -0.31 | $2 \times 10^{-3}$ | -0.37 | $2 \times 10^{-4}$ |
| 2015 W | 0.19 | 0.02 | -0.19 | 0.02 |
| 2016 W | 0.22 | $3 \times 10^{-3}$ | -0.26 | $5 \times 10^{-4}$ |



## 3.4 Gap-filled annual cycles

Cumulative CH$_4$ emissions based on different gap-filling methods are presented in Figure 7. All
follow a similar annual curve, with a steeper increase in summer, but also relatively high
wintertime contribution. Annual, wintertime, and unfrozen period emissions by all gap-filling
methods, with their estimated uncertainties, are shown in Figure 8. Emission estimation by each
sector and data gap-filled by the different method are presented in Table S3. Average values from
all models with their upper and lower limit and wintertime contribution to fluxes are
demonstrated in Table 7.

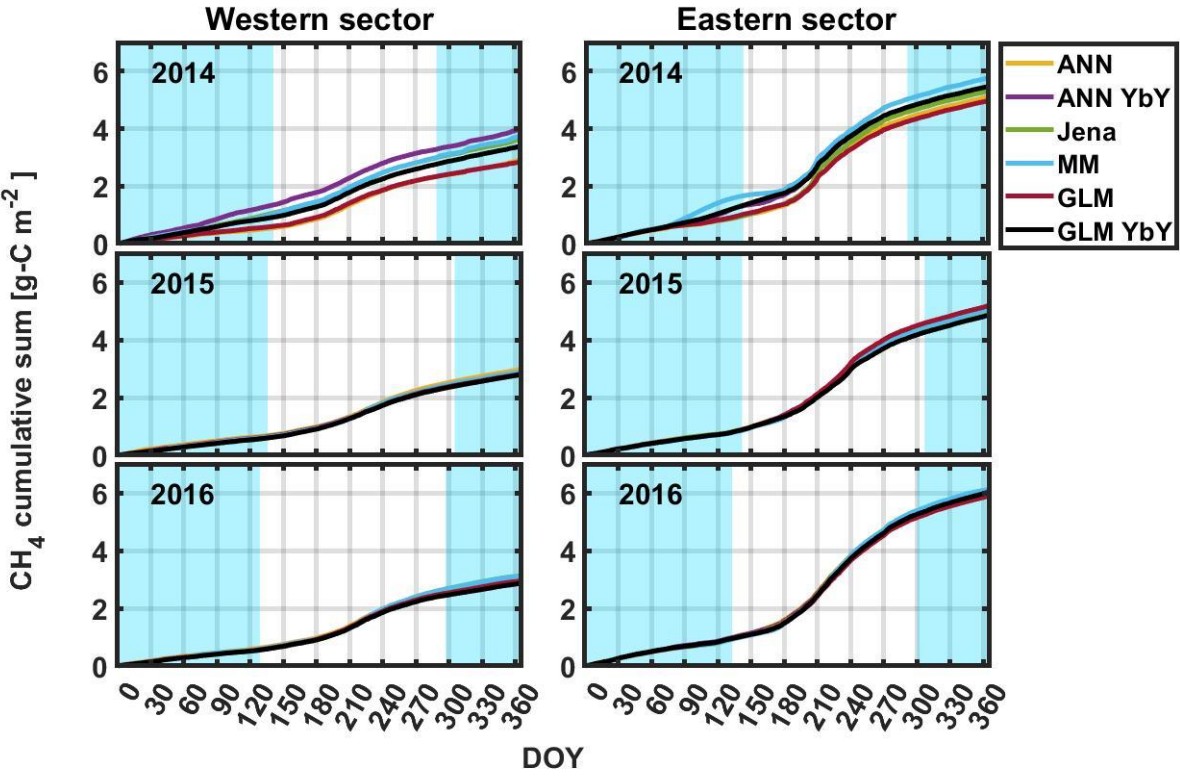

526

Figure 7. The cumulative sum of CH$_4$ fluxes for the years 2014-2016 for western and eastern sectors
calculated with the different gap-filling methods. ANN - the artificial neural network for all years, ANN
YbY - artificial neural network each year separately, Jena - Jena online gap-filling tool, MM - moving
mean with 5-day moving window, GLM- the general linear model for all years, GLM YbY - the general
linear model for each year separately. The shaded light blue area designates the frozen period when
peat temperature at 10 cm was below 0 °C (see Section 2.8 for a detailed description).

533

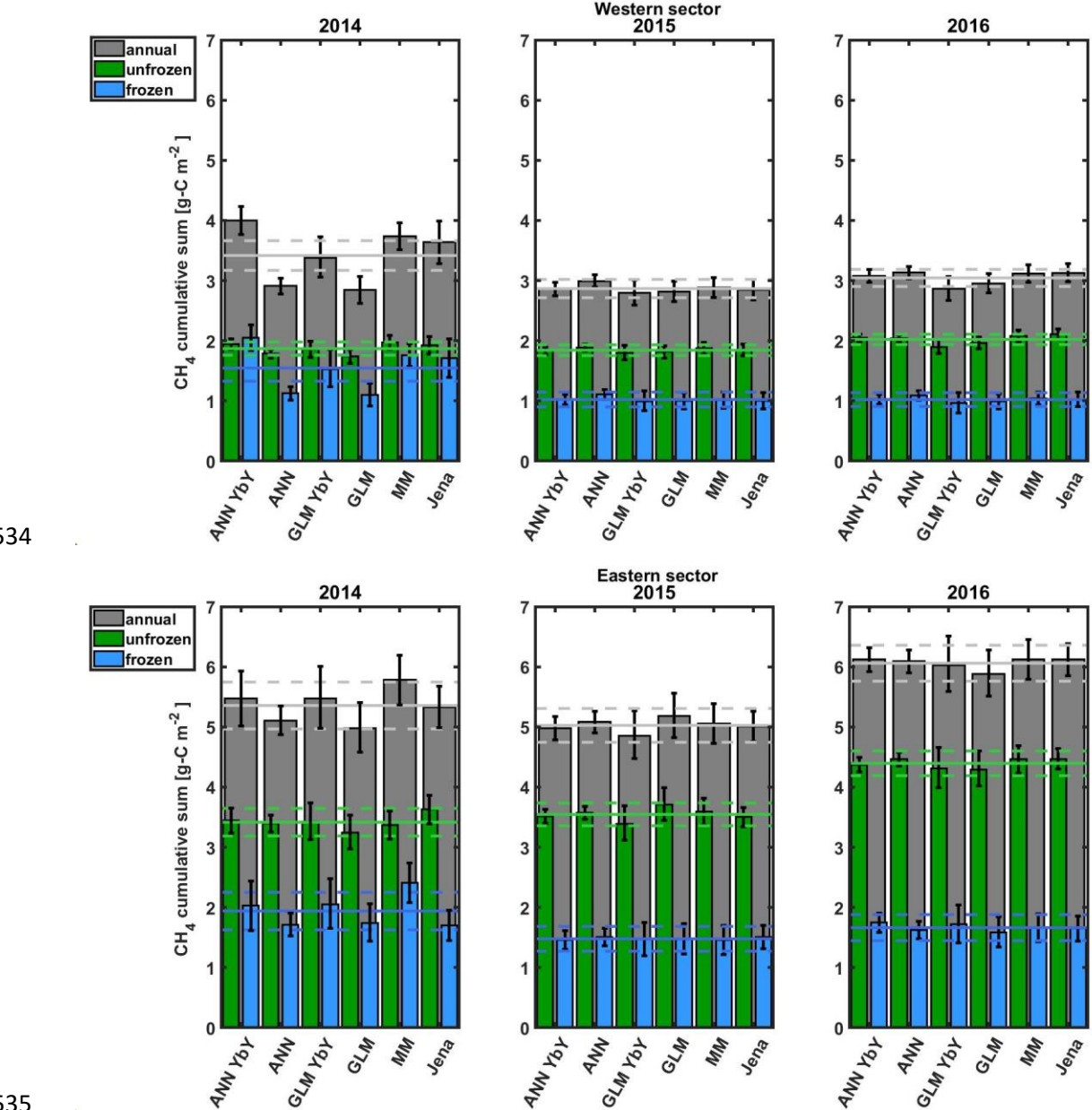

534

535

Figure 8. Comparison of cumulative sums of CH$_4$ fluxes for different gap-filling methods for the western sector (top panel) and eastern sector (bottom panel). ANN - the artificial neural network for all years, ANN YbY - artificial neural network each year separately, Jena - Jena online gap-filling tool, MM - moving mean with 5-day moving window, GLM- the general linear model for all years, GLM YbY - the general linear model for each year separately. Gray bars are for the annual sums, blue bars are for the frozen period sums and green bars are for the unfrozen period (see Section 2.8 for a detailed description). Solid lines are the mean value from all models and dashed lines are for the standard deviation range, with the same colors described above.

As can be seen in Table 3, the year 2014, with a larger difference between annual emissions calculated by different gap-filling methods, had very long gaps that were not present in other years. Also, the uncertainties in annual emission are the largest for the year 2014 for all gap-filling methods, reflecting the gap distribution.

Table 7. Average $CH_4$ annual emission based on all models with the upper and lower limit and contribution from the winter fluxes.

| | Mean | Lower limit | Upper limit | Contribution to wintertime fluxes | Mean | Lower limit | Upper limit | Contribution to wintertime fluxes |
|---|---|---|---|---|---|---|---|---|
| | | | Western sector | | | | Eastern sector | |
| | | g-C m$^{-2}$ a$^{-1}$ | | % | | g-C m$^{-2}$ a$^{-1}$ | | % |
| 2014 | 3.4 | 2.8 | 4.0 | 45 | 5.4 | 5.0 | 5.8 | 36 |
| 2015 | 2.8 | 2.8 | 3.0 | 36 | 5.0 | 4.9 | 5.2 | 29 |
| 2016 | 3.1 | 2.8 | 3.1 | 34 | 6.1 | 5.9 | 6.1 | 27 |

Three years' averages of GPP and net ecosystem exchange (NEE) for two sectors are presented in table 8. As comparison, data from lake and tall sedge fen areas at the Stodalen mire complex, where permafrost was completely thawed, are also presented (Jammet et al., 2017). The fen has the highest percentage of carbon emitted as $CH_4$, as compared to the annual $CO_2$ uptake. The eastern and the western sectors emitted less of the assimilated carbon as $CH_4$ compared to the completely thawed area. The uptake of carbon as $CO_2$ was also largest at the fen.

Table 8. Average annual GPP, NEE and $CH_4$ emission from western and eastern sector in comparison to fen.

| | GPP | NEE | $CH_4$ | $CH_4$/GPP | $CH_4$/NEE |
|---|---|---|---|---|---|
| | g-C m$^{-2}$ a$^{-1}$ | g-C m$^{-2}$ a$^{-1}$ | g-C m$^{-2}$ a$^{-1}$ | % | % |
| Western sector | 225 | -28.9 | 3.1 | 1.4 | 19.6 |
| Eastern Sector | 257 | -42.0 | 5.5 | 2.2 | 14.0 |
| Fen (Jammet et al. 2017) | N.A. | -66.3 | 21.2 | N.A. | 32.0 |

The 3 years' annual average $CH_4$ emissions of palsa and thawing surfaces, as calculated by Eq. (1) and (2), are presented in Table 9. For comparison average annual emissions from other major surface types, measured by EC technique, are shown as well. The emission from the tall

graminoid fen, a third mire type common at Stordalen Mire, has been previously measured using the EC method by Jackowicz-Korczyński et al. (2010) and Jammet et al. (2017). In addition to these, the mire complex includes shallow lakes. Their annual $CH_4$ emission has been measured by EC method by Jammet et al., (2017).

Table 9. Annual $CH_4$ emission from different components of the Stordalen Mire complex from EC studies.

| type of wetland | Annual emission [g-C $m^{-2}$ $a^{-1}$] | References |
|---|---|---|
| palsa plateau surface | 2.7 ± 0.5 | this study |
| thawing wet surface | 8.2 ± 1.5 | this study |
| thawed fen | 15.8 ± 1.6 | Jackowicz-Korczyński et al. 2010 |
| thawed fen | 21.2 ± 1.3 | Jammet et al. 2017 |
| shallow lake | 4.9 ± 0.6 | Jammet et al. 2017 |

## 4 Discussion

### 4.1 Differences in controlling factors

According to the GLM, peat temperature and GPP were typically the first parameters selected by the algorithm to explain $CH_4$ fluxes. In the eastern sector, the $CH_4$ flux correlated best with the peat temperature at 30 cm depth, and in the western sector with the peat temperature at 10 cm depth. Temperature as a controlling factor of $CH_4$ emission has been reported in many wetlands studies (Christensen et al. 2003, Jackowicz-Korczyński et al. 2010, Bansal et al. 2016, Pugh et al. 2017, Rinne et al. 2007; 2018), in line with our findings. The correlation of $CH_4$ fluxes with the temperature at 5 cm depth was also higher than for 30 cm in the western sector. As the peat in the palsa is frozen at 30 cm depth for most of the growing season, the correlation between $CH_4$ fluxes and temperature at these depths is lower. Temperature correlation for the upper part, 2 cm, and 5 cm depth, shows a similar level of correlation as presented by Jackowicz-Korczyński et al. (2010). As they did not analyze correlation with the temperature at deeper peat, we cannot compare these results. The hysteresis-like behavior of the $CH_4$ flux – temperature relation is similar to that observed by Chang et al. (2020) when using temperatures measured above the depth of maximum correlation, but inversed when using temperatures measured at deeper depths (Figure 6). This is in line with at least part of the hysteresis-like behavior to be due to the lag of seasonal temperature wave at the depth of $CH_4$ production compared to the timing of the temperature wave at shallower depth or air temperature.

GPP was indicated as a controlling factor for $CH_4$ emission from a boreal fen ecosystem by Rinne et al. (2018). In our study, the correlation matrix shows a significant correlation between daily average GPP and $CH_4$ flux at both sectors (Table S3). To disentangle the confounding effects of temperature and GPP, we used temperature-normalized $CH_4$ fluxes following Rinne et al. (2018) which revealed that the correlation between GPP and temperature-normalized $CH_4$ flux was not significant in most years (Table 6). Only the data from the eastern sector in the year 2016 shows a significant correlation. Thus, it seems hard to disentangle the effects of temperature and GPP

on CH$_4$ fluxes using this data set. As our data set consists of only three years, the analysis of
interannual variations would not be a robust approach either.
Solar shortwave incoming radiation was selected as a controlling variable by 6 of 8 GLM models
(Table S3). This parameter has an indirect effect on CH$_4$ production via photosynthesis and
subsequent substrate production. The maximum emission of CH$_4$ occurs later in the year than
maximum radiation. This may be due to the CH$_4$ emission depending on the deeper peat
temperature or seasonal cycle of available substrates, lagging behind the annual cycle of
radiation (e.g. Rinne et al., 2018; Chang et al., 2020). The negative contribution of shortwave
radiation in GLM can be due to the slight diel cycle of CH4 emission, with lowest values at
daytime. Mechanistically we can think that the solar irradiance will heat the top of the peat layer,
thus leading to increased methanotrophy at daytime (see discussion above on diel cycle). This
can lead to situation where the methanotrophy is higher in sunny days with warm surface and
lower in cloudy days. The role of photosynthesis for the substrate supply of methanogenesis is
likely to act in the seasonal time scale, where its effect can be masked by the strong correlation
between peat temperature and CH4 emission. The highest correlation of CH$_4$ flux and radiation
was observed in 2014, but GLM did not select radiation as an explanatory factor for this year.
Other years and the whole period show a much lower correlation.
CH$_4$ fluxes from wetlands have been shown to depend on WTL in many studies (e.g. Bubier et al.,
2005; Turetsky et al., 2014; Rinne et al., 2020). However, in a number of studies, the CH$_4$ fluxes
have shown to be relatively insensitive to the small variation, without strong extreme conditions,
in the WTL (Rinne et al. 2007, 2018, Jackowicz-Korczyński et al. 2010). In the eastern sector, CH$_4$
flux and WTL were correlated for the years 2014 and 2016. However, after normalization of CH$_4$
fluxes with their temperature dependence following Rinne et al., (2007), correlations were
mostly not significant (Table 6). This is similar to conclusions drawn by e.g. Rinne et al. (2007,
2018) and Jackowicz-Korczyński et al. (2010).
Instead of WTL, we used SWC as a possible controlling factor for the CH$_4$ emission from the
western sector. Sturtevant et al. (2012) also reported SWC as a controlling factor in autumn. SWC
shows correlation on a significant level before and after normalization for three years for the
western sectors (Table 6).
The GLM algorithm selected SWC as one of the explaining factors while constructing the GLM for
the eastern sector for the whole measurement season. It was chosen by models built for three
years together and each year separately. R and p-value are presented in Table 6. A reduction of
R and increase in p-value after temperature normalization is similar to previous parameters. The
correlation of CH$_4$ emission with SWC stays on a significant level only in the year 2016.

## 4.2 Gap-filling methods
In general, the gap-filled annual CH$_4$ emissions were within their estimated uncertainty from each
other, apart from the year 2014. The results of different gap-filling methods were affected by the
different gap distributions and lengths in different years and the two wind sectors. Thus, below
we discuss the method performance separately for the year 2014 and the two other years.
The dataset from the eastern sector was gap-filled with higher confidence than for the western
sector in 2014. The data from the eastern sector contains fewer very long gaps - more than 30
days, and fewer long gaps - more than 8 but less than 30 days. The method which was most
affected by long gaps was the moving mean approach, indicating that this method should not be
used for data sets with very long gaps. The ANN and the GLM gap-filling methods based on the
whole data set estimated lower annual emission than mean emission from all methods. For two
years without very long gaps (2015 and 2016), the Jena gap-filling tool was assumed as a baseline
method, as it is commonly used for gap-filling of especially $CO_2$ fluxes.  It is independent of the
user choices, as the ecosystem variables required have been chosen by the developers. However,
as this gap-filling tool has been developed for $CO_2$, not all the variables are necessarily relevant
for the gap-filling of the $CH_4$ time series. Furthermore, the Jena gap-filling tool works in a half-
hourly resolution to resolve the diel variation in $CO_2$ fluxes. As the sub-daily variation in $CH_4$ fluxes
is largely random noise in many mires (Rinne et al., 2007; 2018; Jackowicz-Korczyński et al., 2010),
developing a similar tool working at daily time step for $CH_4$, and with tailored parameter set for
$CH_4$, would be useful.
The moving mean approach resulted in annual fluxes within the range of standard deviation from
the Jena gap-filling tool. Daily values probably vary less than values obtained by the Jena tool
because moving means smooth the data. Additional advantages of this method are low input
requirements, as no auxiliary data is needed.
Annual estimates of $CH_4$ emission, based on the gap-filling with algorithms developed for the
whole data set, could be biased when the ecosystem is changing fast between the years and
functional dependencies on environmental parameters change. The annual $CH_4$ emissions by
ANN, based on the whole data set and based on one-year data, agree within the standard
deviation for the years 2015 and 2016. Both of them are also in agreement with the baseline
method within the standard deviation.
The feasibility of GLM is similar to ANN. The GLM model built on the whole dataset is sensitive to
rapid changes in ecosystem functioning and the number of gaps each year. A year with more gaps
has a lower influence on the model, similarly to the ANN. However, annual $CH_4$ emissions derived
using GLMs, based on each year separately or the whole dataset, agree with one another and
with baseline model within the standard deviation. GLM required more preparation than ANN.
Before developing the GLMs, highly correlated parameters need to be determined. The selection
of relevant variables is crucial for the correct performance of that algorithm and the selection
influences model output and model uncertainties.
According to the analysis with artificial gaps, the 35-day artificial gap did not change annual sums
significantly for any gap-filling method. The 80-day artificial gap created a significant difference
for the eastern sector in the year 2015 for ANN YbY and 2016 for ANN (Figure S7). The unfrozen

period did not show significant differences between annual sums for any method. The wintertime period was statistically different for the year 2015 for ANN YbY. The results with the 80-day gap had higher uncertainties than the results with a 35-day gap. The existence of gaps in the winter period did not have a significant impact on the unfrozen period fluxes.

All presented methods show similar $CH_4$ emissions. Choosing one of them as the most appropriate is not obvious, because all of them show both advantages and disadvantages. The method that required the least amount of preparation before use and that was thus the fastest to apply is the moving mean. It can be used for short gaps with good results and does not need additional measured variables to work properly. The ANN method require less preparation than other methods i.e. following the template or choosing the correct variables and it gives similar results. It could be recommended as a gap-filling method suitable for different sites due to unique construction of the ANN for each place.

## 4.3 Winter fluxes

The winter fluxes from both sectors were positive, which is in line with observations by e.g. Rinne et al. (2007, 2018, 2020) and Jammet et al. (2017) of wintertime $CH_4$ emissions from frozen northern mires. Winter emission and potential spring thaw bursts of $CH_4$ can be mechanistically connected (Taylor et al. 2018), while degassing of $CH_4$ during the winter is likely to lead to smaller or no thaw bursts of $CH_4$. Thus, EC studies on the seasonal cycle of $CH_4$ emissions from other seasonally frozen mire ecosystems have shown minor or no thaw emission pulse (Rinne et al., 2007; 2018; Mikhaylov et al. 2015). On the contrary, many studies show spring-thaw emissions from shallow lakes (Raz-Yaseef et al. 2017, Jammet et al. 2015, 2017). In lakes, winter fluxes can be blocked by a solid ice layer leading to the build-up of $CH_4$ below ice during the frozen period (Jammet et al. 2017). On mires, however, the ice cover is not as solid as in lakes, but more porous due to peat and plants within the ice. Therefore, the diffusion during the frozen period is considerably faster than through lake ice. Furthermore, Song et al. (2012) showed that spring burst events could occur at a very small scale and very short in duration (e.g. 2 hours). Small-scale events show a lower influence on EC measurements because the method averages over a larger area. Moreover, if the small-scale short-duration event does not happen in the EC footprint e.g. due to wind direction, it will be missed.

We did not observe an autumn freeze-in burst in our data from either sector at Stordalen Mire. These events have been observed at a High-Arctic tundra site (Mastepanov et al. 2013) though not every year. Mastepanov et al. (2008) suggested that freeze-in bursts of $CH_4$ could be observed only in the Arctic with continuous permafrost and not in a subarctic area with discontinuous or sporadic permafrost. The phenomenon is assumed to be connected to the expansion of water upon freezing, causing air bubbles to be mechanically pushed out of the freezing soil.

## 4.4 Different permafrost status and $CH_4$ emissions

Stordalen Mire is a complex mire system, with at least three different main wetlands surface types and different permafrost status within a distance of a few hundred meters. The permafrost palsa development and thaw depend both on temperature and snow cover and it is partly self-regulating via the effect of microtopography on local snow depth (Johansson et al. 2006). Due to the recently increasing temperatures, the thaw processes are currently likely to dominate over palsa growth. $CH_4$ emission from the different microforms in mire systems depends on the hydrological and nutrient status and temperature which affect e.g. plant and microbial communities.

The carbon emitted as the $CH_4$ fluxes from the eastern and western sector is on similar level to the Siikaneva fen (Rinne et al. 2018). In comparison to the other fen sites reviewed by Rinne et al. (2018), the ratio of $CH_4$ to NEE at Stordalen Mire is higher. The reason behind this could be the shorter growing season and thus lower $CO_2$ fluxes.

The average annual $CH_4$ emissions from different surfaces (Table 9) shows that the palsas have the lowest annual $CH_4$ emissions, followed by a lake. The fully thawed fen, dominated by tall graminoids, has very high annual $CH_4$ emissions and the highest of the mire complex, surpassing e.g. many boreal poor fens (Nilsson et al., 2008; Rinne et al., 2018). The thawing surfaces common in the eastern footprint of the tower have annual $CH_4$ emissions between palsas and tall sedge fen. The three surface types studied here and previously by Jackowicz-Korczyński et al. (2010) and Jammet et al., (2017) can be seen as forming a thaw gradient in this subarctic environment. The globally rising temperature is likely to lead to continuing permafrost thaw in this kind of ecosystem and increased $CH_4$ emissions.

## 5 Conclusion

At our study site, eddy covariance fluxes were measured for two different subarctic mire areas, one dominated by palsa plateaus and the other a mixture of palsas and thawing wet surfaces. The measurements revealed clear differences in their annual $CH_4$ emissions, with the area dominated by palsas emitting less. The annual emission from a thawing surface ($8.2$ g-C m$^{-2}$ a$^{-1}$) was nearly three times higher than from palsa surfaces ($2.7$ g-C m$^{-2}$ a$^{-1}$) but only half of the emission previously reported from fully thawed tall graminoid fen. Areas measured in this study had similar seasonal cycles of emission, with maxima appearing in August and lower but significant fluxes in winter. The seasonal cycles were furthermore characterized by a fast increase in spring (average $0.21$ mg-C m$^{-2}$ d$^{-2}$ for the western sector and $0.68$ mg-C m$^{-2}$ d$^{-2}$ for the eastern sector) and a less rapid decrease in fall (average $-0.16$ mg-C m$^{-2}$ d$^{-2}$ for the western sector and $-0.37$ mg-C m$^{-2}$ d$^{-2}$ for the eastern sector), without any obvious burst events during spring thaw or autumn freeze-in. The wintertime period (from January to mid-May and from late-October to December) contributed with 27 % - 45 % to the annual emission.

According to the correlation matrix and GLM analysis, $CH_4$ emissions from the western and
eastern sectors were partly controlled by different factors. As in most studies on $CH_4$ emission
from wetlands, peat temperature was the most important factor explaining the emission. The
relation of $CH_4$ flux with peat temperature at shallower depths showed similar hysteresis-like
behavior than observed by Chang et al. (2020), but inverse behavior with temperature at deeper
peat. We showed that the existence and direction of hysteresis-like behavior can depend on
which depth the temperature is measured.
The correlation of $CH_4$ emission and WTL in the eastern sector was not significant, but in the
western sector, the SWC did appear to control the emission.
The estimation of annual $CH_4$ emission was based on gap-filling with four different methods. All
methods resulted in similar annual fluxes, especially for the two years with just relatively short
gaps (less than 8 days). The performance of the methods was also dependent on the gap
distribution. Long gaps (more than 8 days) were the most problematic to be reconstructed by
any of the methods. The average annual emission from the western sector was 3.1 g-C $m^{-2}$ $a^{-1}$
and from the eastern sector was 5.5 g-C $m^{-2}$ $a^{-1}$. Both were substantially lower than those
obtained from a tall graminoid fen at the same mire system.
Based on the presented results further studies should focus on winter fluxes, which are important
in the northern, low emissions wetlands with discontinuous permafrost. There is still a lack in
understanding the processes behind those emissions. Also, the origin of wintertime $CH_4$ emission
is somewhat unknown. On the one hand, $CH_4$ can be produced during the winter period, on the
other hand $CH_4$ can also be produced during the growing season, remain stored in the peat and
then be slowly released during the frozen period.  These processes could possibly explain the
hysteresis-like behavior of $CH_4$ emissions.

Data and code availability
http://doi.org/10.5281/zenodo.4640164s

Author contribution
P.Ł., J.H. T.F., P.C. and J.R. analysed and interpreted the data. P.Ł., J.H., P.C., J.R. wrote the
manuscript. T.F., P.C. and, N.R. designed the measurements. N.K. was responsible for the
footprint calculation and its interpretation. P.-O.O. and L.E. were responsible for interpreting
UAV data. A. P. supported with the water table level data.

Competing interests
The authors declare that they have no conflict of interest

Acknowledgements
This study is funded by MEthane goes Mobile: MEasurement and MOdeling (MEMO2) project
from the European Union's Horizon 2020 research and innovation programme under the Marie
Sklodowska-Curie grant agreement No 722479. Data was provided by the Abisko Scientific
Research Station (ANS) and Swedish Infrastructure for Ecosystem Sciences (SITES, co-financed by
the Swedish Research Council) hosting the Stordalen site, part of the ICOS-Sweden network
which was co-financed by the Swedish Research Council (grant-no. 2015-06020, 2019-00205).
Image collection using the UAV was done by Matthias Siewert in collaboration with the SITES
Spectral project.

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
