# Peer review of "Patryk Łakomiec1, Jutta Holst1, Thomas Friborg2, Patrick Crill3, Niklas Rakos4, Natascha Kljun5, Per-2 Ola Olsson1, Lars Eklundh1, Andreas Persson1, Janne Rinne1 3 4 1 Department"

_Biogeosciences, 2021_

## Author Comment (AC1)

We like to thank the reviewer for the time spent and the valuable comments. They helped us to improve the manuscript. Please, see below our answers to the single comments.

Reviewer 1:

*1) in Figure 2 WTL (water depth) is expressed in m a.s.l. It should be presented in relation to the ground level. In line 164 authors report that the EC system collects data from a height of 2.2 m a.g.l., which means that the ground level is somehow determined. The WTL should be related to this level to provide information if the WTL is above ground level or at a certain depth in the ground.*

WTL is related now to the ground level. Figure 2 has been updated and shown below. Furthermore, we found out that the WTL data from 2014 had unknown offset, as the elevation of the sensor was not properly recorded in the metadata. We have now calibrated the WTL data against another dataset, to remove this offset..

[Figure]

Figure1. Figure2 from the manuscript with the calibrated WTL

*2) Figure 2 shows a jump in WTL between 2014 and the next two years by about 1 m for the western sector and several dozen cm (> 50 cm) for the eastern sector. It is surprising that the formation of such a thick peat aeration layer has not affected the CH4 flux.*

After calibration the WTL, jump is not seen any more. Western sector is drier than the eastern sector.

*3) In the case of such a WTL jump, its distribution should be bimodal (why in Figure S2 it is clearly visible for the eastern sector, and not visible for the western sector?) and the correlation between CH4 flux and WTL should be checked separately for each year.*

WTL was not representative for the palsa part of the mire so correlation between CH4 flux and WTL has not been checked for this region. For the eastern sector this correlation was checked separately for each year and results are presented in table7. After calibration correlation between CH4 and WTL was checked one more time. Values are presented here and will update the table7.

Table1. Correlation and p-value for the calibrated WTL

| Year and ecosystem | R for CH$_4$ flux | the p-value for CH$_4$ flux | R for temperature normalized CH$_4$ flux | the p-value for temperature normalized CH$_4$ flux |
|---|---|---|---|---|
| 2014 E | -0.5 | $2x10^{-4}$ | $1x10^{-2}$ | 0.94 |
| 2015 E | -0.2 | 0.3 | -0.2 | 0.17 |
| 2016 E | 0.6 | $4x10^{-6}$ | -0.3 | 0.01 |

*4) Even for the eastern sector, for which WTL is considered representative (ln. 380), the soil moisture in summer 2014 was lower than in the following two summers, while WTL was much higher in 2014 (Figure 2). How is it possible?*

Calibrated WTL data and soil moisture are consistent now. Year 2014 was drier than other two years and it is visible in the WTL and soil moisture (figure above)

*Ln. 397: Please mark the contribution level (e.g. 80%, 50%) on at least a few selected lines in Fig.3.*

We will add% markers to the isolines.

*Ln. 405: As seen in Figure S4, no diel cycle was observed – in my opinion Fig. S4 shows weak diel cycle, with 10-20% differences between nighttime and noon fluxes. Moreover, the potential diel cycle should be exanimated separately in the seasons. In summer, changes in solar radiation can cause a significant diel cycle of surface temperature*

(temperature impulse), which may affect methanogenesis, while in winter there is no such forcing.

Figure S4 shows indeed slightly lower emission during mid-day hours. However, the difference is very small compared to the short term variation in the fluxes as indicated by the interquartile range. Thus, for the purpose of gapfilling this effect could be negligible in calculating daily averages. However, it is interesting to observe this type of diel cycle, with minima at daytime, and as reviewer suggested, it could have its origins on temperature cycle of the top peat layer. This could affect the methanotrophy, while the methanogenesis occurring at slightly deeper layers would be less affected. This would lead to higher methanotrophy at daytime and thus lower emission. We will include this discussion to the revised version of the manuscript.

*Ln. 410-414: Information in Tab. 3 are a bit misleading. For example, for 2016 the coverage by a good data is 99% (sum for eastern and western sector), which seems quite unrealistic for EC method. In fact, the assumption that 10 good data over a full 24 hours is sufficient to calculate daily value (ln. 408) is a kind of gap-filling method and means that up to 58% (14/24) of data might be gap-filled by mean daily value.*

The data point here refers to daily average value, which is indeed kind of gapfilling method. We will make this clearer in the caption of the table. As we require 10 half-hourly flux values (that have passed QC/QA) within a day, we can have days when we have acceptable daily average value for both wind sectors. The percentages for east and west are thus not additive. We can this have many days in 2016 that do not have average methane flux value for either sector or have it for both (59 points are common gaps or values). Additionally, model comparison does not show significant differences in CH4 annual emission based on the 30 minutes ("Jena" method) and daily averages (other methods). If the daily cycle and averaging with only 10 data points had significant impact, it would be visible in the figure 6 or figure 7.

*Ln. 430-431: The peak season of the CH4 emission was defined as two weeks forward and backward from the day with the maximum daily emission in a given year– it is possible that a single high emission does not occur in the peak of the season, so why not use a 14-day moving average and next use the maximum of this function as the peak emission?*

Differences in estimated day between maximum emission and moving average were not big. We changed growing season emission according to the new calculated period. Below we presented estimated DOY and Table1. Day with the highest emission calculated from the daily data and as the maximum of the moving mean

Table2. Middle of the peak season calculated in the two different methods

|        | Non avg | mov avg |
|--------|---------|---------|
|        | DOY     |         |
| 2014 E | 210     | 208     |
| 2015 E | 240     | 245     |
| 2016 E | 221     | 220     |

| | | |
|---|---|---|
| 2014 W | 204 | 210 |
| 2015 W | 224 | 226 |
| 2016 W | 218 | 225 |

Table3. Emission from the new calculated peak season

| | Mean | Standard deviation | The standard error of the mean |
|---|---|---|---|
| | [mg-CH4 m-2 d-1] | | |
| 2014 E | 54.3 | 23.0 | 5.8 |
| 2015 E | 45.9 | 15.6 | 4.9 |
| 2016 E | 60.7 | 9.0 | 2.3 |
| 2014 W | 24.9 | 4.3 | 1.1 |
| 2015 W | 21.5 | 4.3 | 1.3 |
| 2016 W | 27.9 | 3.5 | 0.9 |

Table4. Emission from the peak season – old values

| | Mean | Standard deviation | The standard error of the mean |
|---|---|---|---|
| | [mg-CH$_4$ m$^{-2}$ d$^{-1}$] | | |
| 2014 E | 54.2 | 22.3 | 6.5 |
| 2015 E | 55.3 | 13.2 | 2.8 |
| 2016 E | 59.9 | 9.4 | 2.7 |
| 2014 W | 22.6 | 4.5 | 1.2 |
| 2015 W | 21.4 | 4.2 | 1.1 |
| 2016 W | 28.2 | 3.7 | 1 |

*Ln. 436: Wintertime average emissions were 24 mg-CHm-2 d-1 for the eastern sector and 16 mg-CH4 m-2 d-1 for the western sector – but when we compare these values with Fig. 4, the 24 mg-CH4 m-2 d-1 level is clearly above the most of green tringles for wintertime (blue areas). Similarly, the 16 mg-CH4 m-2 d-1 level is above black dots at winter. It means that the quoted average values for the eastern and western sectors are amplified by gap-filled values, i.e., the gap-filled values on average are significantly higher than the measured once. Is that correct? Any reflection on this effect?*

It was calculation error where averages from two winter periods were sum instead of averaging. We will update the values with correct ones in the revised manuscript.

Table6. Winter emission – new values

| | Mean | Standard deviation | The standard error of the mean |
|---|---|---|---|
| | [mg-CH$_4$ m$^{-2}$ d$^{-1}$] | | |
| 2014 E | 12.0 | 3.83 | 0.49 |
| 2015 E | 11.1 | 2.27 | 0.26 |
| 2016 E | 13.1 | 3.48 | 0.35 |
| 2014 W | 9.6 | 3 | 0.64 |
| 2015 W | 7.4 | 1.85 | 0.26 |
| 2016 W | 7.0 | 4.51 | 0.58 |

Table5. Winter emission – previously calculated

| | *Mean* | *Standard deviation* | *The standard error of the mean* |
|---|---|---|---|
| | *[mg-CH$_4$ m$^{-2}$ d$^{-1}$]* | | |
| 2014 E | 24.1 | 5.5 | 0.7 |
| 2015 E | 22.3 | 3.5 | 0.4 |
| 2016 E | 26.3 | 5 | 0.5 |
| 2014 W | 19.2 | 5.2 | 1.1 |
| 2015 W | 14.9 | 2.8 | 0.4 |
| 2016 W | 14.0 | 3.1 | 0.4 |

Figure2. Averaged winter emission marked on the figure4.

[Figure]

*Ln. 450 and Table 7: Controlling factors were examined before and after temperature normalization (Table 7) – please be more specific about which normalization is concerned. The normalization described in lines 402-405 refers to diel cycle. Of course, it doesn't make sense to correlate such normalized values with other (non-normalized) variables. At this point, the authors are likely to use a different normalization (exponential function of temperature), the same as Rinne et al. (2018). However, this only becomes clear on line 559.*

This is correct. We will replace the sentence

"Controlling factors were examined before and after temperature normalization (Table 7), to avoid effect of cross-correlation between explanatory parameters."

with

"Controlling factors were examined before and after temperature normalization of the CH4 fluxes following Rinne et al. (2018) (Table 7). It was done to avoid effect of cross-correlation between explanatory parameters."

*Ln.521-523: …the fen has the highest percentage of carbon emitted as CH4. The eastern and the western sectors emitted less of the carbon as CH4. – these sentences suggests that both ecosystems emit carbon also as CO2, while in the annual scale, they absorb CO2 (and total carbon)*

Here we compare CH4 emission to carbon uptake as CO2. We will reformulate our sentence for clarity. We will replace

"As a comparison, data from a tall sedge fen area, where permafrost was completely thawed, of Stordalen Mire by Jammet et al. (2017) are presented, showing that the fen has the highest percentage of carbon emitted as $CH_4$. The eastern and the western sectors emitted less of the carbon as $CH_4$."

with

"As comparison, data from lake and tall sedge fen areas at the Stodalen mire complex, where permafrost was completely thawed, are also presented (Jammet et al., 2017). The fen has the highest percentage of carbon emitted as $CH_4$, as compared to the annual $CO_2$ uptake. The eastern and the western sectors emitted less of the assimilated carbon as $CH_4$ compared to the completely thawed area. The uptake of carbon as CO2 was also largest at the fen."

*Ln. 576: … small variation, without strong extreme conditions, in the WTL – can WTL changes in >0.5m (differences between 2014 and next two years) be considered small?*

After calibration the WTL was lower in the year 2014, but the difference between two other years is now much smaller. Variation inside one year in the WTL is not extreme.

*Ln. 699-700:The seasonal cycles were furthermore characterized by a gentle increase in spring and a more rapid decrease in fall – in my opinion, Figure 4 does not confirm this, or even suggest something quite the opposite.*

Thank you for this suggestion. We checked it and you are right. Rate of the increase and decrease were estimated and presented in the table below. Line 699-700 will be rewritten.

| Year and ecosystem | spring | autumn |
|---|---|---|
| | [mg-CH4 m-2 d-2] | |
| 2014 E | 1.63 | -0.59 |
| 2015 E | 0.42 | -0.35 |
| 2016 E | 0.69 | -0.55 |
| 2014 W | 0.4 | -0.22 |
| 2015 W | 0.21 | -0.16 |
| 2016 W | 0.24 | -0.25 |

*Ln.: 706-707: the temperature at different depths seemed to control the CH4 fluxes for the two analyzed mire sectors – can the temperature profile measured at one location*

*east of the tower be representative of the entire eastern (patched) sector? Is the temperature at the set depth the same for the entire eastern sector? The same for western sector. So the conclusion seems a bit too firm.*

This is a good point. The temperature profiles especially in the eastern sector are likely to be patchy. However, disentangling the functional relations of different sub-footprint scale patches can prove impossible. As we expect the higher fluxes from wetter patches we choose to let the temperature measured at wet patch to represent the eastern sector. We will reformulate the conclusion to reflect these caveats.

---

## Author Comment (AC2)

We like to thank the reviewer for the time spent and the valuable comments. They helped us to improve the manuscript. Please, see below our answers to the single comments.

*Generally: in l. 58 you first introduce methane (CH4), but later you switch randomly between "CH4" and "methane" in the text. To provide consistency, please use always "CH4" in the text after first mentioning it in l. 58. Please check the same also for other abbreviations you introduced.*

Thank you for this comment, we will unify the use of abbreviations throughout the manuscript.

*l. 99: This sentence might be difficult to understand. I recommend to divide it into two sentences for each first and second area.*

We will divide it from "The first area is dominated by permafrost plateau, while the second one is thawing, wetter areas. " to

"The first area is dominated by drained permafrost plateau. The second area is thawing and thus resulting in wetter conditions.".

*l. 124 and others: there is a space character missing between value and unit. Please write "0 °C" instead of "0°C" and check also the other parts of the manuscript regarding that.*

Thank you for this comment, we will unify space character between values and units.

*l. 126 and later: in many parts of the manuscript you give both air and peat temperatures with 2 decimal places. Is this really justified, considering the uncertainties of the sensors?*

We will change it to 1 decimal place

*l. 153: The intake tube of the LGR analyzer had a length of almost 30 metres, which is a relatively long tubing. Did you carefully check whether the measured CH4 signal was dampened due to the flow characteristics of the sampling tube? How does the co-spectra look like? Are there any signs for a dampening effect in the high-frequency range, and if possible, did you apply a suitable correction? Please provide a short statement on that in your manuscript.*

We analyzed this with the cospectra of the $CH_4$ and w. This does not show a dampening effect at the high frequency (see figure below), thus the high frequency attenuation does not seem to be very large. Furthermore, the postprocessing software we used to calculate fluxes includes correction for high-frequency losses. We will add a statement on this in the text.

[Figure]

*l. 160: The LI-7200 ist an enclosed path analyzer. Additionally, the official notation of the manufacturer is "LI-COR". Please write it consistent in the manuscript.*

In principle enclosing an open path analyzer will make it a closed path analyzer, no matter what term the manufacturer uses. We will however change the text following manufacturers terminology throughout the manuscript.

*In l. 257, 316 you write "global radiation", in l. 465, 467, 565 you name it "shortwave radiation". I recommend to write "shortwave incoming radiation" generally in the entire manuscript*

 We will change to shortwave incoming radiation throughout the manuscript.

*l. 436: You report an average emission of 24 mg-CH4 m-2 d-1 for the eastern sector in wintertime, which is in accordance with Table 6. However, refering to Fig. 4, wintertime emissions at the eastern sector seem to be substantially lower than 24 mg-CH4 m-2 d-1. Are the mean values, maybe, in Table 5 and 6 the gap-filled ones? a) If yes, please clarify in the table descriptions and in l. 433, l. 437. b) If yes, why does the gap-filled value seem to be substantially higher than the the non-gap-filled data? c) If no, what is the reason for this discrepancy?*

It was calculation error where averages from two winter periods were sum instead of averaging. New values were calculated and will be used in the revised manuscript.

|         | Mean  | Standard deviation | The standard error of the mean |
|---------|-------|--------------------|--------------------------------|
|         | [mg-CH$_4$ m$^{-2}$ d$^{-1}$] | | |
| 2014 E  | 12.04 | 3.83               | 0.49                           |
| 2015 E  | 11.08 | 2.27               | 0.26                           |
| 2016 E  | 13.12 | 3.48               | 0.35                           |
| 2014 W  | 9.58  | 3                  | 0.64                           |
| 2015 W  | 7.4   | 1.85               | 0.26                           |
| 2016 W  | 6.98  | 4.51               | 0.58                           |

*l. 637, "Method...": is there a word missing at the beginning of the sentence?*

We will change from "Choosing one of them as the most appropriate is not obvious, because all of them has strong and week points. Method required the less preparation before use, so the faster to apply is moving mean." to

"Choosing one of them as the most appropriate is not obvious, because all of them show both strong and week points. The method that required the least amount of preparation before use and that was thus the fastest to apply is the moving mean.".

*l.699f: You conclude a "gentle increase" of CH4 fluxes in spring, and a "more rapid decrease in fall". Figure 4 somewhat differs to that finding: I see no difference in increase / decrease ratio for 2016, while for 2014 and 2015 there seems to be a more rapid increase in spring, followed by a less rapid decrease in fall? Am I wrong?*

Thank you for this suggestion. We checked it and you are right. Speed of the increase and decrease were estimated and presented in the table below. Line 699-700 will be rewritten.

| Year and ecosystem | spring | autumn |
|--------------------|--------|--------|
|                    | [mg-CH4 m-2 d-2] | |
| 2014 E             | 1.63   | -0.59  |
| 2015 E             | 0.42   | -0.35  |
| 2016 E             | 0.69   | -0.55  |
| 2014 W             | 0.4    | -0.22  |
| 2015 W             | 0.21   | -0.16  |
| 2016 W             | 0.24   | -0.25  |

*Fig. 1: change m/s => m s-1.*

We will change it.

*Fig. 2: The water table level (WTL) is given in metres above sea level. For what reason? I guess it could be more intuitive to give relative values referencing to the ground level. In l. 164 you introduced a ground level (a.g.l.) baseline - maybe you could do that also for WTL?*

We have calibrated WTL based on the different dataset and it is in the m a.g.l. (figure below). Furthermore, we found out that the WTL data from 2014 had unknown offset, as the elevation of the sensor was not properly recorded in the metadata. We have now calibrated the WTL data against another dataset, to remove this offset.

[Figure]

*Fig. 3, upper panel: To avoid misunderstandings, I recommend to add the information that the red contour lines correspond to the 10 % to 90 % contributions of the flux.*

We will add % markers to the isolines.

*Fig. 5: Shouldn't you change "temp" to "surface peat temperature" in the x-axis label? Additionally, you never use the term "breakout week" in neither text nor the figure itself. Please clarify the figure and/or figure description.*

We will change "temp" to the "peat temperature".  Also we will change the text from

"Figure 5. Weekly averages of CH4 fluxes vs surface peat temperature (top panels), vs the best correlated layer (middle panels), and vs the deeper layer (bottom panel). Data were divided into the beginning of the growing season (blue dots) and end of the growing season (orange triangles), where breakout week was the week with the highest emission." to:

"Figure 5. Weekly averages of CH4 fluxes vs surface peat temperature (top panels), vs the best correlated layer (middle panels), and vs the deeper layer (bottom panel). Data were divided into the beginning of the growing season (blue dots) and end of the growing season (orange triangles). Weeks with emission before reaching the maximum weekly averaged emission was defined as the beginning of the growing season. Weeks with emission after reaching the maximum weekly averaged emission was defined as the beginning of the growing season. "

*Table 1: Tables are always harder to understand than figures, especially when comparing*

*different years and footprints. I suggest to replace table 1 by a figure with the DOY (1 - 366) on the x-axis, and the years (2014, 2015, 2016) on the y-axis. You then draw the unfrozen periods into the plot, using the color codes (grey = western, green = eastern) of Fig. 4. This makes it much simpler to compare different years and footprints.*

Write an answer here…

[Figure]

*Table 7: I guess, "normalization" means that you used temperature-based normalization approach following Rinne et al. (2018) as stated in l.559? This makes sense, however it remains somewhat unclear until reading l. 559 later. Please clarify in the table description and in l. 450 to avoid misunderstandings.*

This is correct. We will replace the sentence

"Controlling factors were examined before and after temperature normalization (Table 7), to avoid effect of cross-correlation between explanatory parameters."

with

"Controlling factors were examined before and after temperature normalization of the CH4 fluxes following Rinne et al. (2018) (Table 7). It was done to avoid effect of cross-correlation between explanatory parameters."

*Table 9: Please write the unit "g-C m-2 yr-1" to be consistent with Table 8 and Table 10.*

We will change it to the one unit throughout the manuscript.

*Table 10: The annual emissions of the type "thawing wet surface" is 11 +/- 2 g-CH4 m-2 yr-1? Do you mean +/- 2.0? Additionally: why is the cell in first column, fifth row, which refers to 28.3 +/- 1.7 g CH4 m-2 yr-1 empty? It is also "thawed fen", I guess?*

Yes, the two values were for the thawed fen.

| type of wetland | Annual emission [$g\text{-}CH_4\ m^{-2}\ yr^{-1}$] | References |
|---|---|---|
| palsa plateau | 3.6 ± 0.7 | this study |
| thawing wet surface | 11.0 ± 2.0 | this study |
| thawed fen | 21.1 ± 2.2 | Jackowicz-Korczyński et al. 2010 |
| thawed fen | 28.3 ± 1.7 | Jammet et al. 2017 |
| shallow lake | 6.5 ± 0.8 | Jammet et al. 2017 |

---

## Author Comment (AC3)

We like to thank the reviewer for the time spent and the valuable comments. They will help us to improve the manuscript. Please, see below our answers to the single comments.

Reviewer 3.

*General:*

*I think that the introduction and discussion are a bit narrow when it comes to referencing related work. The manuscript mostly refers to studies previously performed at Stordalen mire, however, there is not much discussion about related work around the Arctic.*

Thank you for bringing this up. We will add other studies, not directly related to Abisko-Stordalen to the discussion.

Specific comments:

*l. 62: Do you refer here to surface-near air temperature or soil temperature or permafrost temperature?*

We referred to the near-surface air temperature. We will specify this in the text of revised version.

*l. 104-109: Pleae refer here in the introduction to previous studies that have conducted similar comparisons, e.g., Hommeltenberg et al. (2014), Rößger et al. (2019), Kim et al. (2020). Rößger et al. (2020) investigated methane fluxes from a heterogenous tundra ecosystem; thus this article would be quite appropriate for comparison also in other regards*

Thank you for the suggestions. We will add those papers to the introduction

*l. 123: Specify if surface-near air temperature is meant.*

We will clarify that air temperature was meant. We will replace the sentence "The mean annual temperature in this region has been increasing… " with "The mean annual air temperature in this region has been increasing… "

l. 153: The tube length is very long. Can you assure that flow was turbulent throughout the tube? What ist he high-frequency attenuation of the fluxes due to the tube transport effects?

We analyzed this with the cospectra of the $CH_4$ and w. This does not show a dampening effect at the high frequency (see figure below), thus the high frequency attenuation does not seem to be very large. Furthermore, the postprocessing software we used to calculate fluxes includes correction for high-frequency losses We will add a statement on this in the text.

[Figure]

*l. 168-181: Please describe better the locations of the ancillary soil measurements. In a heterogenous mire landscape peat temperature can have large spatial variability. Particularly of interest is what site you choose as being representative for the heterogeneous eastern area composed of drier palsas and thawed wetter sites.*

Soil pit temperature and moisture probe for western sector is located on a palsa plateau. However, the WTL probe is located in a pond approximately 10 m away from the soil temperature and soil moisture sensors, as there is no water table above the permafrost of palsas. Soil pit temperature and moisture probe for the eastern sector is located in the wet area. WTL probe is located in the wetter area located approximately 10 m away.

*l. 202-203: I do not understand this approach of removing flux values when two consecutive data points originated from different wind direction sectors? Which flux values where then removed? Why was this done?*

We will change the text from "Also, flux values when two consecutive data points originated from different wind direction sectors were removed." to

"Also, consecutive data points originating from the two pre-defined wind direction sectors were removed to avoid influences from unstationary conditions".

*l. 213-215: Have you tried to model also 30 min fluxes? Why not modelling the 30 min flux data (as Rößger et al. (2019))?*

We have not tried 30 minutes' gap-filling by ANN, but did so with Jena-tool. We portioned data in the different way that Rößger et al. (2019). In the comparison of the model, results from the model based on the 30 minutes ("Jena") does not show any better performance than models

designed on the daily averages. As most of the sub-daily variation of CH4 fluxes seems to be due to random turbulent variability, modelling this variability may not be very fruitful, at least for gapfilling purposes.

*l. 222-223: Please describe in more detail how the 30 min data were „aggregated" to annual footprint climatologies.*

We followed the standard approach to derive footprint climatologies (e.g. Kljun et al. 2015), where footprint function values are aggregated for each grid cell (50 cm x 50 cm). We will add more detailed description to the revised manuscript.

*l. 235-237: How is the weighting calculation exactly done? Which quantity of the „climatology" was used for weighting the contribution of a mosaic pixel?*

Again, we refer to Kljun et al. (2015) where the derivation of a footprint climatology and its application to remote sensing data is described. The flux contribution per pixel is weighted with the footprint function value.

*l. 245-246: The statement that "...methane emissions ... do not show diel cycle" is too bold. Figure S4 shows that there is systematic diurnal variability – even for whole-year data. Indeed, it would be good to analyse diurnal variability month by month.*

Figure S4 shows indeed slightly lower emission during mid-day hours. However, the difference is very small compared to the short term variation in the fluxes as indicated by the interquartile range. Thus, for the purpose of gapfilling this effect could be negligible in calculating daily averages. However, it is interesting to observe this type of diel cycle, with minima at daytime, and as reviewer suggested, it could have its origins on temperature cycle of the top peat layer. This could affect the methanotrophy, while the methanogenesis occurring at slightly deeper layers would be less affected. This would lead to higher methanotrophy at daytime and thus lower emission. We will include this discussion to the revised version of the manuscript

*l. 274: Rößger et al. (2019) applied ANN for a heterogenous tundra; the paper might be interesting for comparison*

Thank you for the reference. We will incorporate it in the manuscript.

*l. 323-333: I think that the equations (1) and (2) are only valid under rather strong assumptions that should be clearly stated. In my view equation (1) and equation (2) can be considered valid for the 30-min periods for which the footprint contributions of the two contrasting landcover types fp and ft are estimated. However, using the same form of the equation for annual averages is only valid if the time series of the footprint contributions fp and ft, respectively, are uncorrelated with the temporal development of the emission factors Ep and Et, respectively. If they would show some correlation, the average of the product f\*E would equal the product of the averages of f and E, respectively, plus the covariance of f and E. Therefore, one maybe important assumption is that f and E of the*

*respective landcover types are uncorrelated. Other important assumptions are that the average methane fluxes of the palsa sites in the eastern area and in the western area are equal and that the average methane fluxes of the thawed sites in the eastern area and in the western area are equal. It would be good if this assumption could be backed by more comprehensive description of microtopography, hydrology and vegetation of palsa and thawed sites of the western and eastern areas, respectively.*

Yes, we will clearly state the assumptions in the revised version of the manuscript. Of course this is estimate has its caveats but we feel it is useful as an attempt to separate the effect of different surfaces to aggregate flux. Figure below shows monthly averages of the contribution of surface classes to the footprint. According to the figure relative contribution of the different surface classes is stable during each year and also between years.

[Figure]

Figure1. Relative contribution of the different surface classes to footprint

*l. 360-361: Sentence too vague: Where and when the western sector is colder? Which depth? Time scale?*

We will change the text from "The western sector, was colder than the eastern sector, causing the existence of permafrost. " to

"During our investigation period (2014-2016),the peat temperature from 30 to 50 cm below ground was colder in the western sector, than those of the eastern sector, corresponding to the existence of the permafrost."

*l. 369, Figure 2: The lowest panel does not show water depth but water table height. However, it would be more suitable to show water table depth or height referenced to a reference point at the ground surface in the investigated mire. The jump in water table*

*height between the summer of 2014 and the summer of 2015 appears unrealistic. Please check the water level times series for biases and inhomogeneities in the time series.*

The sensors which measure WTL are taken out every autumn to prevent damage through freezing. Unfortunately, the exact installation depth is not completely documented for 2014, only showing that it is not comparable to the following years. The jump seen in WTL is thus artificial. The reference point at that year is different than in the other years, and unfortunately there is no record of it. However, there is another two-year data set (2014-2015) we can use to introduce a correction factor to WTL data from 2014. We will make this change to our data and re-run the analysis with WTL. After correction the WTL data is in line with soil moisture data, indicating the 2014 to be the driest year (Figure below).

[Figure]

Figure1: Figure 2 of the manuscript, with corrected WTL data.

*l. 387-390: Please write more specific, e.g. "…on average over all three years more than 90% to the fluxes measured at the eddy covariance tower."*

You need to write at least "We replace…", or something, to begin the answer.

"Footprint and flux contribution of drier and wetter areas are presented in Figure 3. The dry areas (yellow) contribute to more than 90 % fluxes from the western sector. In the eastern sector, the wetter (blue) and drier areas contribute almost equally to the fluxes. The contributions of the wet and dry areas to the fluxes in both sectors were stable across the three study years" to.

"Footprint and flux contribution of drier and wetter areas are presented in Figure 3. The dry areas (yellow) contribute *on average over all three years more than 90% to the fluxes measured at the eddy covariance tower* from the western sector. In the eastern sector, the wetter (blue) and drier areas contribute almost equally to the fluxes. The contributions of the wet and dry areas to the fluxes in both sectors were stable across the three study years"

*l.392, Caption figure 3: Please describe more precise what is shown in the bottom panel. How are these average contributions of contrasting landcover types calculated. Generally, I would prefer another diagram type that allows evaluation of the variability of footprint contributions of the two landcover types.*

We prefer to retain the figure as it is now. Variability of the different land covers is quite stable during the whole measurement period as presented on the figure above. We will add the figure on the variability to supplementary material, and refer to it in the revised manuscript.

*l. 419-420: How did you deal with the autocorrelation in the time series? Serial dependence of data points could lead to biased results of the Wilcoxon test.*

Thank you for this comment. Autocorrelation existed up to 8 days. With this information I divided winter data to the subsets where every 9th day was taken. We tested the difference of those subsets without autocorrelation to zero with Wilcoxon rank sum test. Tests made in this way also rejected hypothesis that winter fluxes are equal to zero.

*l. 455: Unclear what "breakout week" means*

We will change the text from

"Figure 5. Weekly averages of CH4 fluxes vs surface peat temperature (top panels), vs the best correlated layer (middle panels), and vs the deeper layer (bottom panel). Data were divided into the beginning of the growing season (blue dots) and end of the growing season (orange triangles), where breakout week was the week with the highest emission." to:

"Figure 5. Weekly averages of CH4 fluxes vs surface peat temperature (top panels), vs the best correlated layer (middle panels), and vs the deeper layer (bottom panel). Data were divided into the beginning of the growing season (blue dots) and end of the growing season (orange triangles). Weeks with emission before reaching the maximum weekly averaged emission was defined as the beginning of the growing season. Weeks with emission after reaching the maximum weekly averaged emission was defined as the beginning of the growing season. "

*l. 565-572: I find the discussion of the explanatory strength of incoming shortwave radiation confusing. The GLM parameters in Table S2 for the explanatory variable incoming shortwave radiation are all negative, indicating that methane emissions were lowered under high incoming shortwave radiation. Thus, the GLM results do not suggest strong relations between shortwave radiation, photosynthesis, substrate supply and CH4 production. Or was the sign convention for incoming radiation different than I assumed?*

Thank you for the comment. We will rewrite this part of the discussion. The negative contribution of shortwave radiation can be due to the slight diel cycle of CH4 emission, with

lowest values at daytime. Mechanistically we can think that the solar irradiance will heat the top of the peat layer, thus leading to increased methanotrophy at daytime (see discussion above on diel cycle). We can lead to situation where the methanotrophy is higher in sunny days with warm surface and lower in cloudy days. The role of photosynthesis for the substrate supply of methanogenesis is likely to act in the seasonal time scale, where its effect can be masked by the strong correlation between peat temperature and CH4 emission.

[Figure]

*l. 724-725: The hysteresis-like behaviour can be also explained by the phase lags between different temperatures in air, ground surface and different soil depths*

This is actually our point. In the papers by Chang et al., (2020; 2021) hysteresis with air and shallow peat temperature are shown, and as explanation, precursor availability is used. We want to show that the existence and direction of hysteresis-like behavior can depend on which depth the temperature is measured. As reviewer writes, the phase lag between the temperatures at different depths is likely explanation, at least partially.

*Figure S1: Specify in the caption if soil or air temperature is shown. At which height in the atmosphere?*

We will specify this in the text.

*Technical:*

We will apply all technical comments.

---

## Author Response (AR1)

Referee #1 comments

We like to thank the reviewer for the time spent and the valuable comments. They helped us to improve the manuscript. Please, see below our answers to each comment.

*1) in Figure 2 WTL (water depth) is expressed in m a.s.l. It should be presented in relation to the ground level. In line 164 authors report that the EC system collects data from a height of 2.2 m a.g.l., which means that the ground level is somehow determined. The WTL should be related to this level to provide information if the WTL is above ground level or at a certain depth in the ground.*

WTL is related now to ground level. The revised Figure 2 is shown below. Furthermore, we found out that the WTL data from 2014 had an unknown offset, as the elevation of the sensor was not properly recorded in the metadata. The original WTL data has now been corrected against an independent dataset, to correct this offset.

[Figure]

Figure1. Revised Figure 2 for the manuscript, with the corrected WTL

*2) Figure 2 shows a jump in WTL between 2014 and the next two years by about 1 m for the western sector and several dozen cm (> 50 cm) for the eastern sector. It is surprising that the formation of such a thick peat aeration layer has not affected the CH4 flux.*

The jump was caused by the wrong offset applied to the data in 2014. It is not present in the

data after the correction of the WTL. With the revised WTL data the western sector was consistently drier than the eastern sector.

*3) In the case of such a WTL jump, its distribution should be bimodal (why in Figure S2 it is clearly visible for the eastern sector, and not visible for the western sector?) and the correlation between CH4 flux and WTL should be checked separately for each year.*

For the eastern sector the correlation between CH4 flux and WTL was checked separately for each year (Table 7 in the manuscript). The correlation results were updated after the WTL offset correction in the manuscript (Table 7) and are presented below (Table 1). WTL for the palsa part was not measured in the nearer pond so it is not fully represented for the whole western area correlation between $CH_4$ flux and WTL has not been checked for this region. More representative value was SWC. However, figure1 shows similar pattern for the SWC and WTL.

Table 1. Correlation and p-value for the calibrated WTL

| Year and ecosystem | R for CH$_4$ flux | the p-value for CH$_4$ flux | R for temperature normalized CH$_4$ flux | the p-value for temperature normalized CH$_4$ flux |
|---|---|---|---|---|
| 2014 E | -0.5 | $2 \times 10^{-4}$ | $1 \times 10^{-2}$ | 0.94 |
| 2015 E | -0.2 | 0.3 | -0.2 | 0.17 |
| 2016 E | 0.6 | $4 \times 10^{-6}$ | -0.3 | 0.01 |

*4) Even for the eastern sector, for which WTL is considered representative (ln. 380), the soil moisture in summer 2014 was lower than in the following two summers, while WTL was much higher in 2014 (Figure 2). How is it possible?*

Offset corrected WTL data and soil moisture are consistent now. Year 2014 was drier than other two years and it is visible in the WTL and soil moisture (figure above)

*Ln. 397: Please mark the contribution level (e.g. 80%, 50%) on at least a few selected lines in Fig.3.*

We added % markers to the isolines. The revised Figure is shown below.

[Figure]

*Ln. 405: As seen in Figure S4, no diel cycle was observed – in my opinion Fig. S4 shows weak diel cycle, with 10-20% differences between nighttime and noon fluxes. Moreover, the potential diel cycle should be exanimated separately in the seasons. In summer, changes in solar radiation can cause a significant diel cycle of surface temperature (temperature impulse), which may affect methanogenesis, while in winter there is no such forcing.*

We changed "As seen in Figure S4 no diel cycle was observed" to "Figure S5 shows slightly lower emission during mid-day hours. However, the difference is small compared to the short-term variation in the fluxes as indicated by the interquartile range. Thus, for the purpose of gap-filling this effect could be negligible in calculating daily averages. However, it is interesting to observe this type of diel cycle, with minima at daytime. It could have its origins on temperature cycle of the top peat layer. This could affect the methanotrophy, while the methanogenesis occurring at slightly deeper layers would be less affected. This would lead to higher methanotrophy at daytime and thus lower emission. "

*Ln. 410-414: Information in Tab. 3 are a bit misleading. For example, for 2016 the coverage by a good data is 99% (sum for eastern and western sector), which seems quite unrealistic for EC method. In fact, the assumption that 10 good data over a full 24 hours is sufficient to calculate daily value (ln. 408) is a kind of gap-filling method and means that up to 58% (14/24) of data might be gap-filled by mean daily value.*

The data point here refers to daily average value, which is indeed a kind of gap-filling method.

As we required 10 half-hourly flux values (that have passed QC/QA) within a day, we can have days when we have acceptable daily average value for both wind sectors. The percentages for east and west are thus not additive. Due to this, we can have days in 2016 that do not have average methane flux value for either sector or have it for both (59 points are common gaps or values). Additionally, model comparison does not show significant differences between annual $CH_4$ emission based on the 30 minute averages ("Jena" method) and that based on daily averages (other methods). If the daily cycle and averaging with only 10 data points had significant impact, it would be visible in the figure 6 or figure 7. We changed the caption of the table from "Table 3. The size of available daily data sets after averages for each year and wind sector" to "Table 3. The size of available daily data sets after gap-filling by daily averaging for each year and wind sector."

*Ln. 430-431: The peak season of the CH4 emission was defined as two weeks forward and backward from the day with the maximum daily emission in a given year– it is possible that a single high emission does not occur in the peak of the season, so why not use a 14-day moving average and next use the maximum of this function as the peak emission?*

We changed the definition of the peak season according to your valuable suggestion. New definition of the peak season "The center point of the peak season of the $CH_4$ emission was defined as the day of maximum of the 14-days moving average CH4 emission. Two weeks forward and backward from the center point was defined as the peak season, and peak season emissions were estimated for this period in each year".  Data in the tables and the text have been adjusted accordingly. We changed data in table from:

Table4. Emission from the peak season

| | Mean | Standard deviation | The standard error of the mean |
|---|---|---|---|
| | [mg-CH$_4$ m$^{-2}$ d$^{-1}$] | | |
| 2014 E | 54.2 | 22.3 | 6.5 |
| 2015 E | 55.3 | 13.2 | 2.8 |
| 2016 E | 59.9 | 9.4 | 2.7 |
| 2014 W | 22.6 | 4.5 | 1.2 |
| 2015 W | 21.4 | 4.2 | 1.1 |
| 2016 W | 28.2 | 3.7 | 1 |

To:

Table 5. CH$_4$ emission during the peak season

| | Mean | Standard deviation | The standard error of the mean |
|---|---|---|---|
| | | [mg-C m$^{-2}$ d$^{-1}$] | |
| 2014 E | 40.7 | 17.2 | 4.3 |
| 2015 E | 34.4 | 11.7 | 3.7 |
| 2016 E | 45.4 | 6.7 | 1.7 |
| 2014 W | 18.6 | 3.2 | 0.8 |
| 2015 W | 16.1 | 3.2 | 1.0 |
| 2016 W | 20.9 | 2.6 | 0.7 |

*Ln. 436: Wintertime average emissions were 24 mg-CHm-2 d-1 for the eastern sector and 16 mg-CH4 m-2 d-1 for the western sector – but when we compare these values with Fig. 4, the 24 mg-CH4 m-2 d-1 level is clearly above the most of green tringles for wintertime (blue areas). Similarly, the 16 mg-CH4 m-2 d-1 level is above black dots at winter. It means that the quoted average values for the eastern and western sectors are amplified by gap-filled values, i.e., the gap-filled values on average are significantly higher than the measured once. Is that correct? Any reflection on this effect?*

The effect was a result of a calculation error where data from two winter periods were displayed as sum instead of the mean value. We corrected the values in the table in the revised manuscript.

Table 6. Winter emission

| | Mean | Standard deviation | The standard error of the mean |
|---|---|---|---|
| | | [mg-C m$^{-2}$ d$^{-1}$] | |
| 2014 E | 9.0 | 2.8 | 0.4 |
| 2015 E | 8.3 | 1.7 | 0.2 |
| 2016 E | 9.8 | 2.6 | 0.3 |
| 2014 W | 7.2 | 2.2 | 0.4 |
| 2015 W | 5.5 | 1.4 | 0.2 |
| 2016 W | 5.2 | 3.4 | 0.4 |

*Ln. 450 and Table 7: Controlling factors were examined before and after temperature normalization (Table 7) – please be more specific about which normalization is concerned. The normalization described in lines 402-405 refers to diel cycle. Of course, it doesn't*

*make sense to correlate such normalized values with other (non-normalized) variables. At this point, the authors are likely to use a different normalization (exponential function of temperature), the same as Rinne et al. (2018). However, this only becomes clear on line 559.*

This is correct. We replaced the sentence:

"Controlling factors were examined before and after temperature normalization (Table 7), to avoid effect of cross-correlation between explanatory parameters."

with

"Controlling factors were examined before and after temperature normalization of the $CH_4$ fluxes following Rinne et al. (2018) (Table 7). It was done to avoid effect of cross-correlation between explanatory parameters."

*Ln.521-523: …the fen has the highest percentage of carbon emitted as CH4. The eastern and the western sectors emitted less of the carbon as CH4. – these sentences suggests that both ecosystems emit carbon also as CO2, while in the annual scale, they absorb CO2 (and total carbon)*

Here we compared $CH_4$ emission to carbon uptake as $CO_2$. We reformulated our sentence for clarity. We replaced

"As a comparison, data from a tall sedge fen area, where permafrost was completely thawed, of Stordalen Mire by Jammet et al. (2017) are presented, showing that the fen has the highest percentage of carbon emitted as $CH_4$. The eastern and the western sectors emitted less of the carbon as $CH_4$."

with

"As comparison, data by Jammet et al. (2017) from lake and tall sedge fen areas at the Stordalen mire complex, where permafrost was completely thawed, are also presented. The fen has the highest percentage of carbon emitted as $CH_4$, as compared to the annual $CO_2$ uptake. The eastern and the western sectors emitted less of the assimilated carbon as $CH_4$ compared to the completely thawed area. The uptake of carbon as CO2 was also largest at the fen."

*Ln. 576: … small variation, without strong extreme conditions, in the WTL – can WTL changes in >0.5m (differences between 2014 and next two years) be considered small?*

After offset correction, the WTL was lower in the year 2014, but the difference between two other years is now much smaller. Variation inside one year in the WTL was not extreme.

*Ln. 699-700:The seasonal cycles were furthermore characterized by a gentle increase in spring and a more rapid decrease in fall – in my opinion, Figure 4 does not confirm this, or*

*even suggest something quite the opposite.*

The sentences were changed from "The seasonal cycles were furthermore characterized by a gentle increase in spring and a more rapid decrease in fall, without any obvious burst events during spring thaw or autumn freeze-in." to

"The seasonal cycles were furthermore characterized by a fast increase in spring (average 0.21 mg-C $m^{-2}$ $d^{-2}$ for the western sector and 0.68 mg-C $m^{-2}$ $d^{-2}$ for the eastern sector) and a less rapid decrease in fall (average -0.16 mg-C $m^{-2}$ $d^{-2}$ for the western sector and -0.37 mg-C $m^{-2}$ $d^{-2}$ for the eastern sector), without any obvious burst events during spring thaw or autumn freeze-in."

*Ln.: 706-707: the temperature at different depths seemed to control the CH4 fluxes for the two analyzed mire sectors – can the temperature profile measured at one location east of the tower be representative of the entire eastern (patched) sector? Is the temperature at the set depth the same for the entire eastern sector? The same for western sector. So the conclusion seems a bit too firm.*

We deleted the sentence "However, the temperature at different depths seemed to control the $CH_4$ fluxes for the two analyzed mire sectors." to make conclusion less firm

Referee #2 comments

We like to thank the reviewer for the time spent and the valuable comments. They helped us to improve the manuscript. Please, see below our answers to each comment.

*Generally: in l. 58 you first introduce methane (CH4), but later you switch randomly between "CH4" and "methane" in the text. To provide consistency, please use always "CH4" in the text after first mentioning it in l. 58. Please check the same also for other abbreviations you introduced.*

We have unified the use of abbreviations throughout the revised manuscript.

*l. 99: This sentence might be difficult to understand. I recommend to divide it into two sentences for each first and second area.*

We divided it from "The first area is dominated by permafrost plateau, while the second one is thawing, wetter areas. " to

"The first area is dominated by a drained permafrost plateau. The second area is thawing and thus resulting in wetter conditions.".

*l. 124 and others: there is a space character missing between value and unit. Please write "0 °C" instead of "0°C" and check also the other parts of the manuscript regarding that.*

We have unified the spacing between values and units in the revised manuscript.

*l. 126 and later: in many parts of the manuscript you give both air and peat temperatures with 2 decimal places. Is this really justified, considering the uncertainties of the sensors?*

We changed it to 1 decimal place throughout the revised manuscript.

*l. 153: The intake tube of the LGR analyzer had a length of almost 30 metres, which is a relatively long tubing. Did you carefully check whether the measured CH4 signal was dampened due to the flow characteristics of the sampling tube? How does the co-spectra look like? Are there any signs for a dampening effect in the high-frequency range, and if possible, did you apply a suitable correction? Please provide a short statement on that in your manuscript.*

We added following sentences:

We analyzed this with the co-spectra of the $CH_4$ and the vertical wind speed w. The analysis did not show a dampening effect at the high frequencies (Figure S2), thus the high frequency attenuation does not seem to be very large. Furthermore, the post-processing software we

used to calculate fluxes includes correction for high-frequency losses

[Figure]

Figure S2. Example of the averaged and model co-spectra for the stable and unstable conditions during growing season year 2015

*l. 160: The LI-7200 ist an enclosed path analyzer. Additionally, the official notation of the manufacturer is "LI-COR". Please write it consistent in the manuscript.*

We changed the notation of the manufacturer to the "LI-COR" throughout the manuscript. The LI-7200 is essentially a closed path analyzer, albeit with a short intake tubing, no matter what LI-COR as a company calls it. Thus we wish to retain the nomenclature.

*In l. 257, 316 you write "global radiation", in l. 465, 467, 565 you name it "shortwave*

*radiation". I recommend to write "shortwave incoming radiation" generally in the entire manuscript*

We changed to shortwave incoming radiation throughout the revised manuscript.

*l. 436: You report an average emission of 24 mg-CH4 m-2 d-1 for the eastern sector in wintertime, which is in accordance with Table 6. However, refering to Fig. 4, wintertime emissions at the eastern sector seem to be substantially lower than 24 mg-CH4 m-2 d-1. Are the mean values, maybe, in Table 5 and 6 the gap-filled ones? a) If yes, please clarify in the table descriptions and in l. 433, l. 437. b) If yes, why does the gap-filled value seem to be substantially higher than the the non-gap-filled data? c) If no, what is the reason for this discrepancy?*

The effect was a result of a calculation error where data from two winter periods were displayed as sum instead of the mean value. We updated the correct values to the table 6 in the revised manuscript.

Table 6. Winter emission

| | *Mean* | *Standard deviation* | *The standard error of the mean* |
|---|---|---|---|
| | *[mg-C m$^{-2}$ d$^{-1}$]* | | |
| 2014 E | 9.0 | 2.8 | 0.4 |
| 2015 E | 8.3 | 1.7 | 0.2 |
| 2016 E | 9.8 | 2.6 | 0.3 |
| 2014 W | 7.2 | 2.2 | 0.4 |
| 2015 W | 5.5 | 1.4 | 0.2 |
| 2016 W | 5.2 | 3.4 | 0.4 |

.

*l. 637, "Method...": is there a word missing at the beginning of the sentence?*

We changed from "Choosing one of them as the most appropriate is not obvious, because all of them has strong and week points. Method required the less preparation before use, so the faster to apply is moving mean." to

"Choosing one of them as the most appropriate is not obvious, because all of them show both strong and week points. The method that required the least amount of preparation before use and that was thus the fastest to apply is the moving mean.".

*l.699f: You conclude a "gentle increase" of CH4 fluxes in spring, and a "more rapid decrease in fall". Figure 4 somewhat differs to that finding: I see no difference in increase*

*/ decrease ratio for 2016, while for 2014 and 2015 there seems to be a more rapid increase in spring, followed by a less rapid decrease in fall? Am I wrong?*

The sentence was changed from "The seasonal cycles were furthermore characterized by a gentle increase in spring and a more rapid decrease in fall, without any obvious burst events during spring thaw or autumn freeze-in." to

"The seasonal cycles were furthermore characterized by a faster increase in spring (average 0.21 mg-C m$^{-2}$ d$^{-2}$ for the western sector and 0.68 mg-C m$^{-2}$ d$^{-2}$ for the eastern sector) and a less rapid decrease in fall (average -0.16 mg-C m$^{-2}$ d$^{-2}$ for the western sector and -0.37 mg-C m$^{-2}$ d$^{-2}$ for the eastern sector), without any obvious burst events during spring thaw or autumn freeze-in"

*Fig. 1: change m/s => m s-1.*

We changed the unit notification.

[Figure]

*Fig. 2: The water table level (WTL) is given in metres above sea level. For what reason? I guess it could be more intuitive to give relative values referencing to the ground level. In l. 164 you introduced a ground level (a.g.l.) baseline - maybe you could do that also for WTL?*

We found out that the WTL data from 2014 had an unknown offset, as the elevation of the sensor was not properly recorded in the metadata. The original WTL data has now been corrected against an independent dataset, to correct this offset. Furthermore, WTL has now been related to ground level. The revised figure is shown below.

[Figure]

*Fig. 3, upper panel: To avoid misunderstandings, I recommend to add the information that the red contour lines correspond to the 10 % to 90 % contributions of the flux.*

We added % markers to the isolines in the revised manuscript.

[Figure]

*Fig. 5: Shouldn't you change "temp" to "surface peat temperature" in the x-axis label? Additionally, you never use the term "breakout week" in neither text nor the figure itself. Please clarify the figure and/or figure description.*

We did change "temp" to "peat temperature" in the revised manuscript.  Also we changed the text from

"Figure 5. Weekly averages of CH4 fluxes vs surface peat temperature (top panels), vs the best correlated layer (middle panels), and vs the deeper layer (bottom panel). Data were divided into the beginning of the growing season (blue dots) and end of the growing season (orange triangles), where breakout week was the week with the highest emission."

 to:

"Figure 5. Weekly averages of CH4 fluxes against the surface peat temperature (top panels), the depth with best correlation (middle panels), and the deeper layer (bottom panel). Data were divided into the beginning of the growing season (blue dots) before the maximum weekly emission, and end of the growing season (orange triangles) after that. "

*Table 1: Tables are always harder to understand than figures, especially when comparing*

*different years and footprints. I suggest to replace table 1 by a figure with the DOY (1 - 366) on the x-axis, and the years (2014, 2015, 2016) on the y-axis. You then draw the unfrozen periods into the plot, using the color codes (grey = western, green = eastern) of Fig. 4. This makes it much simpler to compare different years and footprints.*

We changed the Table 1 to a new figure, Figure 2 in the revised manuscript

Figure 2. Time periods of frozen peat during the years 2014 - 2016 (green and black bars) and peak CH$_4$ emission season (dot with whiskers) for the western sector (green) and the eastern sector (black). (For peak season definition see chapter 3.2)

[Figure]

*Table 7: I guess, "normalization" means that you used temperature-based normalization*

*approach following Rinne et al. (2018) as stated in l.559? This makes sense, however it remains somewhat unclear until reading l. 559 later. Please clarify in the table description and in l. 450 to avoid misunderstandings.*

This is correct. We replaced the sentence

"Controlling factors were examined before and after temperature normalization (Table 7), to avoid effect of cross-correlation between explanatory parameters."

with

"Controlling factors were examined before and after temperature normalization of the CH4 fluxes following Rinne et al. (2018) (Table 7). It was done to avoid effect of cross-correlation between explanatory parameters."

*Table 9: Please write the unit "g-C m-2 yr-1" to be consistent with Table 8 and Table 10.*

We changed g-CH$_4$ to g-C throughout the manuscript.

*Table 10: The annual emissions of the type "thawing wet surface" is 11 +/- 2 g-CH4 m-2 yr-1? Do you mean +/- 2.0? Additionally: why is the cell in first column, fifth row, which refers to 28.3 +/- 1.7 g CH4 m-2 yr-1 empty? It is also "thawed fen", I guess?*

Yes, the two values were for the thawed fen.

| type of wetland | Annual emission [g-C m$^{-2}$ yr$^{-1}$] | References |
|---|---|---|
| palsa plateau surface | 2.7 ± 0.5 | this study |
| thawing wet surface | 8.2 ± 1.5 | this study |
| thawed fen | 15.8 ± 1.6 | Jackowicz-Korczyński et al. 2010 |
| thawed fen | 21.2 ± 1.3 | Jammet et al. 2017 |
| shallow lake | 4.9 ± 0.6 | Jammet et al. 2017 |

Referee #3 comments

We like to thank the reviewer for the time spent and the valuable comments. They will help us to improve the manuscript. Please, see below our answer to each comment.

*General:*

*I think that the introduction and discussion are a bit narrow when it comes to referencing related work. The manuscript mostly refers to studies previously performed at Stordalen mire, however, there is not much discussion about related work around the Arctic.*

Thank you for bringing this up. We did add other studies, not directly related to Abisko-Stordalen to the discussion (Hommeltenberg et al. (2014), Rößger et al. (2019), Kim et al. (2019)).

Specific comments:

*l. 62: Do you refer here to surface-near air temperature or soil temperature or permafrost temperature?*

We referred to the near-surface air temperature. We specified this in the text of the revised manuscript:

"Ecosystems near the annual near-surface air temperature isotherms of 0 °C are vulnerable to permafrost thaw and changes in ecosystem characteristics in a warming climate."

*l. 104-109: Pleae refer here in the introduction to previous studies that have conducted similar comparisons, e.g., Hommeltenberg et al. (2014), Rößger et al. (2019), Kim et al. (2020). Rößger et al. (2020) investigated methane fluxes from a heterogenous tundra ecosystem; thus this article would be quite appropriate for comparison also in other regards*

We added those papers to the introduction "Test of the four methods will decrease the uncertainty in an annual balance estimation (Hommeltenberg et al. (2014), Rößger et al. (2019), Kim et al. (2019))."

*l. 123: Specify if surface-near air temperature is meant.*

We clarified that air temperature was meant. We replaced the sentence "The mean annual temperature…" with "The mean annual near-surface air temperature…"

l. 153: The tube length is very long. Can you assure that flow was turbulent throughout the tube? What ist he high-frequency attenuation of the fluxes due to the tube transport effects?

We added following sentences:

"We analyzed this with the co-spectra of the CH4 and the vertical wind speed 158 w. The analysis did not show a dampening effect at the high frequencies (Figure S2), thus the high 159 frequency attenuation does not seem to be very large. Furthermore, the post-processing 160 software we used to calculate fluxes includes correction for high-frequency losses"

A figure of the co-spectra was added to the supplement materials.

[Figure]

Figure S2. Example of the averaged and model co-spectra for the stable and unstable conditions during growing season year 2015

*l. 168-181: Please describe better the locations of the ancillary soil measurements. In a*

*heterogenous mire landscape peat temperature can have large spatial variability. Particularly of interest is what site you choose as being representative for the heterogeneous eastern area composed of drier palsas and thawed wetter sites.*

We rewrote section 2.3 from "Ancillary measurements are presented in Table S1. The sampling frequency for these parameters was 1 Hz and the collected data were averaged into half-hourly values. Measured variables are divided into two categories: peat/soil parameters, and meteorological parameters. Peat temperatures at each depth, soil heat fluxes, and soil water contents (SWC) were measured at four plots around the EC tower, located towards the four cardinal directions. In further analysis, data just from two of these locations were used (East and West) as these were within the flux footprints areas of the EC tower. The sites for the water table level (WTL) measurements differed from the peat temperature profiles. Furthermore, data for WTL was available only during the unfrozen period, as the probes were removed during the frozen period to avoid damage. The WTL on the western sector was measured in a wet collapse feature, surrounded by drained areas. The palsa areas commonly have no persistent WTL above the permafrost surface. Meteorological variables were measured on a separate mast, placed 10 meters south-west of the flux measurement mast."

to "Ancillary measurements are presented in Table S1. The sampling frequency for these parameters was 1 Hz and the collected data were averaged into half-hourly values. Measured variables are divided into two categories: peat/soil parameters, and meteorological parameters. Peat temperatures at each depth, soil heat fluxes, and soil water contents (SWC) were measured at four locations around the EC tower, located towards the four cardinal directions. In further analysis, data just from two of these locations were used (East and West) as these were within the flux footprints areas of the EC tower. The sites for the water table level (WTL) measurements differed from the peat temperature profiles. Soil pit for temperature and moisture probe in the western sector is located on a palsa plateau. However, the WTL probe is located in a pond approximately 10 m away from the soil temperature and SWC measurement, as there is no WTL above the permafrost of palsas. Soil pit for temperature and SWC probe in the eastern sector is located in the wet thawing area. WTL probe is located in the wetter area approximately 10 m away. Furthermore, data for WTL was available only during the unfrozen period, as the probes were removed for the frozen period to avoid damage. Meteorological variables were measured on a separate mast, placed 10 meters south-west of the flux measurement mast."

*l. 202-203: I do not understand this approach of removing flux values when two consecutive data points originated from different wind direction sectors? Which flux values where then removed? Why was this done?*

We changed the text from "Also, flux values when two consecutive data points originated from different wind direction sectors were removed." to

"Also, consecutive data points originating from both pre-defined wind direction sectors were removed to avoid influences from non-stationary conditions".

*l. 213-215: Have you tried to model also 30 min fluxes? Why not modelling the 30 min flux data (as Rößger et al. (2019))?*

We have not tried 30 minutes' gap-filling by ANN, but did so with the Jena-tool. We portioned the dataset in differently than Rößger et al. (2019). We divided half-hour data to the two datasets and treat it as two independent sets. Rößger et al. (2019) used half-hour dataset and treated it as one time-series. In the comparison of the model, results from the model based on the 30 minutes ("Jena") do not show any better performance than models designed on daily averages. As most of the sub-daily variation of $CH_4$ fluxes seems to be due to random turbulent variability, modelling this variability may not be very fruitful, at least for gap-filling purposes. Any changes were not done.

*l. 222-223: Please describe in more detail how the 30 min data were „aggregated" to annual footprint climatologies.*

We followed the standard approach to derive footprint climatologies (e.g. Kljun et al. 2015), where footprint function values are aggregated for each grid cell (50 cm x 50 cm). We added following sentences ". I.e. the half-hourly footprint function values were aggregated for each land cover grid cell (50 cm x 50 cm) to derive a footprint-weighted flux contribution per pixel."

*l. 235-237: How is the weighting calculation exactly done? Which quantity of the „climatology" was used for weighting the contribution of a mosaic pixel?*

Again, we refer to Kljun et al. (2015) where the derivation of a footprint climatology and its application to remote sensing data is described. The flux contribution per pixel is weighted with the footprint function value.

*l. 245-246: The statement that "...methane emissions … do not show diel cycle" is too bold. Figure S4 shows that there is systematic diurnal variability – even for whole-year data. Indeed, it would be good to analyse diurnal variability month by month.*

We changed "As seen in Figure S4 no diel cycle was observed" to "Figure S5 shows slightly lower emission during mid-day hours. However, the difference is small compared to the short-term variation in the fluxes as indicated by the interquartile range. Thus, for the purpose of gap-filling this effect could be negligible in calculating daily averages. However, it is interesting to observe this type of diel cycle, with minima at daytime. It could have its origins on temperature cycle of the top peat layer. This could affect the methanotrophy, while the methanogenesis occurring at slightly deeper layers would be less affected. This would lead to higher methanotrophy at daytime and thus lower emission."

*l. 274: Rößger et al. (2019) applied ANN for a heterogenous tundra; the paper might be interesting for comparison*

Thank you for the reference. We added this paper "An artificial neural network (ANN) has been successfully applied for gap-filling of $CH_4$ fluxes by e.g. Dengel et al. (2013), Jammet et al. (2015,2017), Knox et al. (2016) *and* Rößger et al. (2019)."

*l. 323-333: I think that the equations (1) and (2) are only valid under rather strong assumptions that should be clearly stated. In my view equation (1) and equation (2) can be considered valid for the 30-min periods for which the footprint contributions of the two contrasting landcover types fp and ft are estimated. However, using the same form of the equation for annual averages is only valid if the time series of the footprint contributions fp and ft, respectively, are uncorrelated with the temporal development of the emission factors Ep and Et, respectively. If they would show some correlation, the average of the product f\*E would equal the product of the averages of f and E, respectively, plus the covariance of f and E. Therefore, one maybe important assumption is that f and E of the respective landcover types are uncorrelated. Other important assumptions are that the average methane fluxes of the palsa sites in the eastern area and in the western area are equal and that the average methane fluxes of the thawed sites in the eastern area and in the western area are equal. It would be good if this assumption could be backed by more comprehensive description of microtopography, hydrology and vegetation of palsa and thawed sites of the western and eastern areas, respectively.*

There are of course caveats in the simple approach to separate the emission rates of the two surface types followed here. The figure below shows monthly averages of the contribution of surface classes to the footprint. According to the figure, the relative contribution of the different surface classes is stable during each year and also between years. This indicates that there should not be a major correlation between seasonal behavior of emission rates and footprint contributions. Thus we have added the following paragraph on the assumptions to the revised manuscript

"Here we assumed that the emission 348 rate from both palsa and thaw surfaces are equal in eastern and western sectors. Furthermore, 349 we must assume that there is no correlation between footprint contribution and seasonally 350 developing emission rate at either surface type. The seasonally constant contributions of the 351 surface types to the footprint indicate that the latter assumption may well be valid (Figure S4)."

[Figure]

Figure1. Monthly mean of the relative contribution of the different surface cover classes to the tower flux footprint

*l. 360-361: Sentence too vague: Where and when the western sector is colder? Which depth? Time scale?*

We changed the text from "The western sector, was colder than the eastern sector, causing the existence of permafrost. " to

"During our investigation period (2014-2016), the peat temperature from 30 cm to 50 cm below ground was colder in the western sector, than those of the eastern sector, corresponding to the existence of the permafrost"

*l. 369, Figure 2: The lowest panel does not show water depth but water table height. However, it would be more suitable to show water table depth or height referenced to a reference point at the ground surface in the investigated mire. The jump in water table height between the summer of 2014 and the summer of 2015 appears unrealistic. Please check the water level times series for biases and inhomogeneities in the time series.*

We found out that the WTL data from 2014 had an unknown offset, as the elevation of the sensor was not properly recorded in the metadata. The original WTL data has now been corrected against an independent dataset, to correct this offset. Furthermore, WTL has now

been related to ground level. The updated figure is shown below.

[Figure]

*l. 387-390: Please write more specific, e.g. "…on average over all three years more than 90% to the fluxes measured at the eddy covariance tower."*

We changed text from:
"Footprint and flux contribution of drier and wetter areas are presented in Figure 3. The dry areas (yellow) contribute to more than 90 % fluxes from the western sector. In the eastern sector, the wetter (blue) and drier areas contribute almost equally to the fluxes. The contributions of the wet and dry areas to the fluxes in both sectors were stable across the three study years" to:

"Footprint and flux contribution of drier and wetter areas are presented in Figure 3. The dry areas (yellow) contribute on average over all three years more than 90% to the fluxes measured at the eddy covariance tower from the western sector. In the eastern sector, the wetter (blue) and drier areas contribute almost equally to the fluxes. The contributions of the wet and dry areas to the fluxes in both sectors were stable across the three study years."

*l.392, Caption figure 3: Please describe more precise what is shown in the bottom panel. How are these average contributions of contrasting landcover types calculated. Generally, I would prefer another diagram type that allows evaluation of the variability of footprint contributions of the two landcover types.*

We prefer to retain the figure as it is now. The variability of the contribution of the different land covers is quite stable during the whole measurement period as presented on the figure above. We added the figure on the variability to supplementary material (figure S4), and referred to it in the revised manuscript.

*l. 419-420: How did you deal with the autocorrelation in the time series? Serial dependence of data points could lead to biased results of the Wilcoxon test.*

We added following information: "Wilcoxon rank sum test need data without autocorrelation. The autocorrelation in the data existed up to 8 days. Based on this we divided winter data to the subsets where every 9th day was selected. We tested the difference of those subsets to zero with Wilcoxon rank sum test. Winter fluxes were statistically different from zero (p < 0.001, two-sided Wilcoxon rank sum test)."

*l. 455: Unclear what "breakout week" means*

We changed the text from

"Figure 5. Weekly averages of $CH_4$ fluxes vs surface peat temperature (top panels), vs the best correlated layer (middle panels), and vs the deeper layer (bottom panel). Data were divided into the beginning of the growing season (blue dots) and end of the growing season (orange triangles), where breakout week was the week with the highest emission." to:

"Figure 6. Weekly averages of CH4 fluxes against the surface peat temperature (top panels), the depth with best correlation (middle panels), and the deeper layer (bottom panel). Data were divided into the 1$^{st}$ part of the growing season (blue dots) before the maximum weekly emission, and 2$^{nd}$ part of the growing season (orange triangles) after that. "

*l. 565-572: I find the discussion of the explanatory strength of incoming shortwave radiation confusing. The GLM parameters in Table S2 for the explanatory variable incoming shortwave radiation are all negative, indicating that methane emissions were lowered under high incoming shortwave radiation. Thus, the GLM results do not suggest strong relations between shortwave radiation, photosynthesis, substrate supply and CH4 production. Or was the sign convention for incoming radiation different than I assumed?*

We added following information: "The negative contribution of shortwave radiation in GLM can be due to the slight diel cycle of $CH_4$ emission, with lowest values at daytime. Mechanistically we can think that the solar irradiance will heat the top of the peat layer, thus leading to increased methanotrophy at daytime (see discussion above on diel cycle). This can lead to a situation where the methanotrophy is higher in sunny days with warm surface and lower in cloudy days. The role of photosynthesis for the substrate supply of methanogenesis is likely to act in the seasonal time scale, where its effect can be masked by the strong correlation between peat temperature and $CH_4$ emission."

*l. 724-725: The hysteresis-like behaviour can be also explained by the phase lags between different temperatures in air, ground surface and different soil depths*

This is what we wanted to express. In the papers by Chang et al., (2020; 2021) hysteresis with air and shallow peat temperature are shown, and as explanation, the precursor availability is used. We showed that the existence and direction of hysteresis-like behavior can depend on which depth the temperature is measured. We agree with your comment that the phase lag between the temperatures at different depths is likely the explanation, at least partially.

*Figure S1: Specify in the caption if soil or air temperature is shown. At which height in the atmosphere?*

We specified this in the text. "Figure S1. Time series of near-surface air temperature measurements recorded by Abisko SMHI (Sveriges Meteorologiska och Hydrologiska Institut) at ANS (Abisko Naturvetenskapliga Station) at 1.5 m a.g.l., 10 km to the east of Stordalen Mire, with years 2014-2016 indicated with orange circles."

*Technical:*

We applied all technical comments.